# A new computational model illuminates the extraordinary eyes of *Phronima*

**Zahra M. Bagheri**[1,2☯]*, **Anna-Lee Jessop**[1,2☯¤]*, **Julian C. Partridge**[2], **Karen J. Osborn**[3,4], **Jan M. Hemmi**[1,2,3]

**1** School of Biological Sciences, The University of Western Australia, Crawley, Western Australia, Australia,
**2** UWA Oceans Institute, The University of Western Australia, Crawley, Western Australia, Australia,
**3** Department of Invertebrate Zoology, Smithsonian National Museum of Natural History, Washington, DC, United States of America, **4** Monterey Bay Aquarium Research Institute, Moss Landing, California, United States of America

☯ These authors contributed equally to this work.
¤ Current address: College of Science, Health, Engineering and Education, Mathematics and Statistics, Murdoch University, Perth, Western Australia, Australia
* zahra.bagheri@uwa.edu.au (ZMB); annie.jessop@murdoch.edu.au (A-LJ)

**Data Availability Statement:** All relevant data are within the manuscript and its Supporting Information files.

**Funding:** Smithsonian 2019 NMNH ADCS Core Research Grant awarded to KJO and JMH.

## Abstract

Vision in the midwater of the open ocean requires animals to perform visual tasks quite unlike those of any other environment. These tasks consist of detecting small, low contrast objects and point sources against a relatively dim and uniform background. Deep-sea animals have evolved many extraordinary visual adaptations to perform these tasks. Linking eye anatomy to specific selective pressures, however, is challenging, not least because of the many difficulties of studying deep-sea animals. Computational modelling of vision, based on detailed morphological reconstructions of animal eyes, along with underwater optics, offers a chance to understand the specific visual capabilities of individual visual systems. Prior to the work presented here, comprehensive models for apposition compound eyes in the mesopelagic, the dominant eye form of crustaceans, were lacking. We adapted a model developed for single-lens eyes and used it to examine how different parameters affect the model's ability to detect point sources and extended objects. This new model also allowed us to examine spatial summation as a means to improve visual performance. Our results identify a trade-off between increased depth range over which eyes function effectively and increased distance at which extended objects can be detected. This trade-off is driven by the size of the ommatidial acceptance angle. We also show that if neighbouring ommatidia have overlapping receptive fields, spatial summation helps with all detection tasks, including the detection of bioluminescent point sources. By applying our model to the apposition compound eyes of *Phronima*, a mesopelagic hyperiid amphipod, we show that the specialisations of the large medial eyes of *Phronima* improve both the detection of point sources and of extended objects. The medial eyes outperformed the lateral eyes at every modelled detection task. We suggest that the small visual field size of *Phronima*'s medial eyes and the strong asymmetry between the medial and lateral eyes reflect *Phronima*'s need for effective vision across a large depth range and its habit of living inside a barrel. The barrel's narrow aperture limits the usefulness of a large visual field and has allowed a strong

Smithsonian national museum of natural history URL: https://naturalhistory.si.edu/ Australian Research Council's Discovery Projects funding scheme, project number DP180100491, awarded to JMH and JCP. Australian research council URL: https://www.arc.gov.au/ The funders had no role in study design, data collection and analysis, decision to publish, or preparation of the manuscript.

**Competing interests:** The authors have declared that no competing interests exist.

asymmetry between the medial and lateral eyes. The model provides a useful tool for future investigations into the visual abilities of apposition compound eyes in the deep sea.

## Author summary

How do animals see the world? This is particularly an interesting question when the animal's eyes look very different from our own, or if they inhabit an environment that is visually very different from ours. Biologists approach this question by seeking to determine not only how animal eyes function but also what selective pressures led to the evolution of their eyes. Understanding the eyes of deep-sea animals is particularly intriguing and more challenging than usual because their visual world is so dramatically different from our own and they are inaccessible and therefore hard to study. Understanding their visual capabilities by behavioural or physiological experiments is at best extremely challenging and often impossible. However, modelling of their visual abilities, by combining knowledge about ocular anatomy with information about the way light propagates in the deep sea, is comparatively tractable. Here we present a computational model that predicts the ability of apposition compound eyes (eyes that are widely found in many arthropod invertebrates) to detect salient visual targets in the deep sea between 200 and 700 m below the surface. We use this model specifically to examine the extraordinary 'double eyes' of the midwater hyperiid amphipod *Phronima* that have perplexed scientists for decades. This allowed us to put forward a new hypothesis about the selective pressures that have led to *Phronima*'s unusual eyes. The predictive model we present here also provides a framework for future assessments of visual performance of apposition compound eyes in other deep-sea animals.

## Introduction

The visual environment of the deep sea is unlike any other [1]. In the upper layers, the mesopelagic zone (200–1000 m), the visual environment comprises a three-dimensional radiance distribution in which the highest radiance is seen when looking upward and the lowest radiance downward [2]. With increasing depth, this radiance distribution becomes dimmer and restricted in its spectral distribution to blue wavelengths, until we reach the bathypelagic zone (below 1000 m) where solar illumination does not reach even in the clearest water [2]. Salient visual targets such as predators, prey, conspecifics, and food items therefore appear against a relatively homogenous background at all depths of the deep sea [3]. In the mesopelagic, these targets appear as dark silhouettes against a lighter background, or, if they bioluminesce, as luminous objects or point sources against a darker background [3]. The ocular anatomies and the extraordinary variation of visual adaptations in mesopelagic animals reflect both their phylogenetic constraints and the unusual selective pressures imposed by their environments. However, linking anatomical observations to specific selective pressures is challenging, particularly in the case of deep-sea animals.

Deep-sea animals are notoriously difficult to study. Their habitat is mostly inaccessible to us, with direct human observational access requiring the use of expensive submersibles and indirect observations relying on remotely operated vehicles or deployed video platforms. Even when we can access the deep sea, it is difficult to see the animals behaving naturally due to the bright lights of the imaging systems [4]. Observations in the lab avoid some of those problems

but create others. For example, many deep-sea animals are fragile and thus easily damaged or killed during collection by traps, nets, or the journey to the surface. Ship-board physiological or behavioural experiments are therefore extremely challenging, not only because of the above collection and maintenance concerns, but also because the animals are sensitive to even small changes in their surroundings, particularly lighting, temperature, and oxygen concentrations [5]. Consequently, most of what we know about deep-sea vision comes from studying the morphology of eyes [6–8]. Morphology provides important descriptive information about an animals' visual system but combining this with computational modelling provides a much more powerful tool to explore visual performance and to make firm predictions about visual performance [9].

To date, a comprehensive computational model of visual performance in the midwater has been developed only for single-lens eyes [9]. For apposition compound eyes, we had relatively simplistic models that exclude important parameters such as the optical properties of water and the integration time of the eye [10,11]. Here, we build on the models developed by Nilsson et al. [9] to predict the distance at which apposition compound eyes are able to detect three different visual targets in the deep sea. We consider dark and luminous extended objects as well as bioluminescent point sources. We also expanded the model to include the effects of transparency of extended objects, and spatial and temporal summation. Spatial summation is used to improve vision in dim light by many animals, including invertebrates, and is therefore an important consideration when predicting visual performance [12]. Many animals in the deep sea use transparency as a form of camouflage [13] and we expect transparency to have a strong effect on detection distances.

Our model provides an opportunity to examine the extraordinary eyes of the midwater hyperiid amphipod *Phronima* (Fig 1). These small mesopelagic crustaceans are predators and parasites, notable for their habit of creating 'barrels' to live in from the tissue of gelatinous zooplankton such as salps or pyrosomes [14,15]. These barrels provide nutrition, buoyancy, and protection, but also impact *Phronima's* visual ecology. *Phronima* species are well-known for their unusual eyes (Fig 1C) and have been the subject of several studies [6,10,11,16–19]. *Phronima* has four morphologically distinct eyes, two small lateral eyes and two exceptionally large medial eyes. The large medial eyes have narrow and overlapping fields of view of approximately 15˚ and the small lateral eyes have large fields of view extending approximately 180˚ [6]. An unusual feature of the medial eyes is that their pseudopupils are large, being approximately eight ommatidia in diameter, indicating that more than 60 ommatidia view the same point in visual space [10]. Land [10] theoretically demonstrated that if these ommatidia spatially summate their signals, the detection of extended objects should be significantly improved. However, he did not fully consider whether this strategy could also improve the detection of bioluminescent point sources.

It has been convincingly suggested that the function of the large, dorsally directed eyes of many mesopelagic animals, that are assumed to or have been shown to point upwards (e.g., the cockeyed squid [20]), is to detect silhouetted targets against downwelling light [21]. At first inspection, the similarity of *Phronima's* large, dorsally directed eyes suggests that this is their function as well. Braun [19] showed that *Phronima's* positive phototaxis is driven by their large medial eyes. This behaviour persists up to a certain brightness, after which *Phronima* becomes negatively phototactic [17,19]. Based on morphological data, Ball [16] suggested that the medial eyes have evolved to provide relatively high spatial resolution at low light levels, while also enhancing *Phronima's* ability to follow an isolume. However, Land [10] showed that the medial eyes exhibit an enormous pseudopupil indicating that the acceptance angles of ommatidia are large and overlapping, which is not conducive to high spatial resolution. Land theorized that if the medial eyes employed spatial summation, their design would aid in

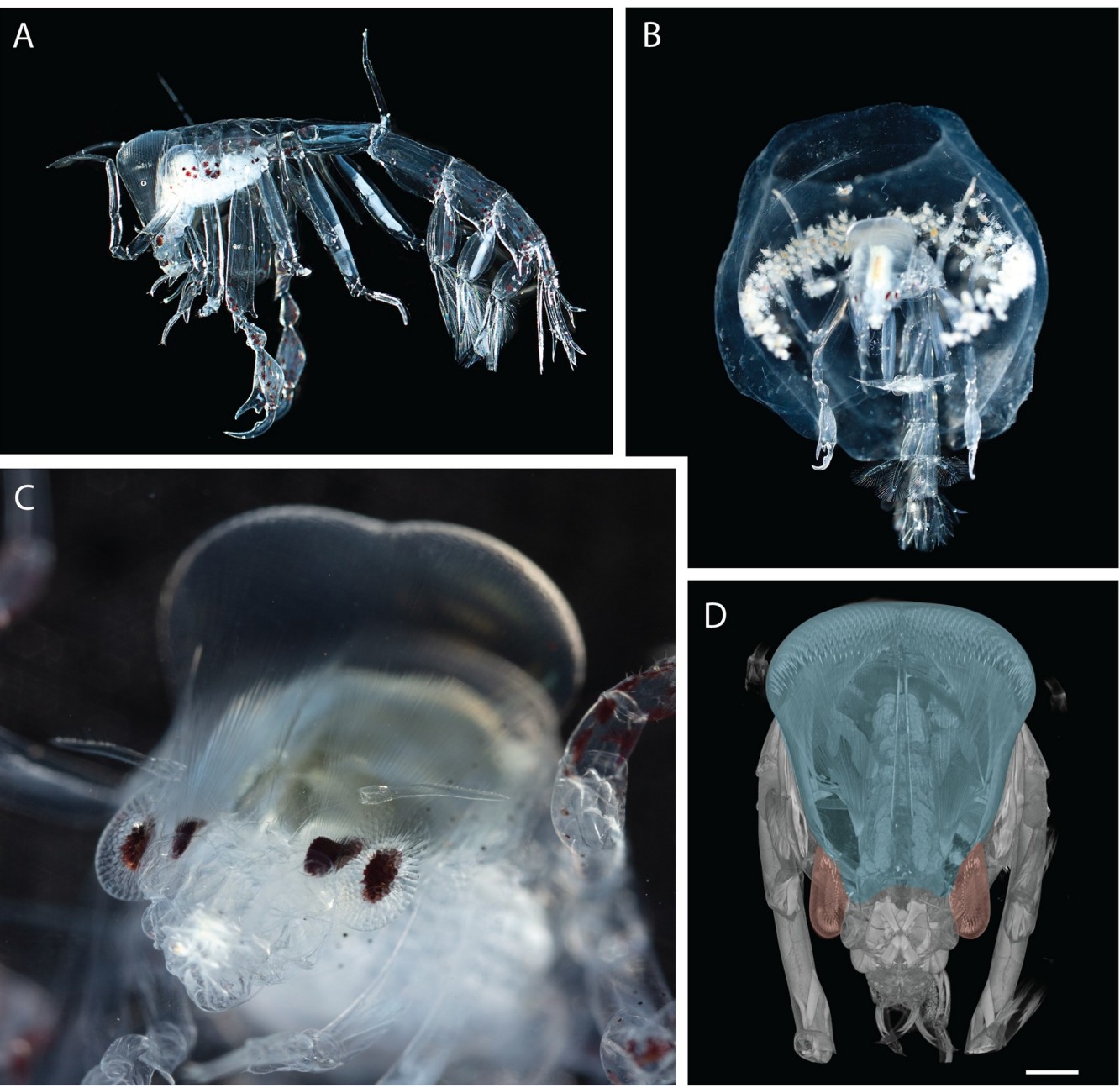

**Fig 1. *Phronima* exemplars.** (A) Lateral view of *Phronima*'s body with head at left. (B) Female *Phronima* in barrel with brood. (C) Close-up of *Phronima*'s eyes with their medial eye retinas visible at the base of their long light guides and the lateral eye retinas visible just lateral to them, closely surrounded by the distinguishable corneal lenses of the ommatidia. The lenses of the medial eyes cover the dorsal surface of the head seen as two bulges at the top of the photo. (D) Three-dimensional reconstructions of the medial and lateral eyes based on micro-computed tomography (micro-CT) image data of Specimen 2. Blue and red shading identify the medial and lateral eyes, respectively. The reconstruction was made using Dragonfly (Object Research Systems, Inc.). Scale bar 1 mm.

detecting small objects in dim light. In contrast, he concluded that the lateral eyes simply provide low resolution all-round vision [10]. Indeed, *Phronima* appeared to track a blue light resembling a bioluminescent point source using only their lateral eyes [18]. Land [11] supported this finding by theoretically demonstrating that the detection of bioluminescent point sources is generally a less demanding task compared to detecting dark extended objects,

requiring only small eyes, like the lateral eyes of *Phronima*. He did point out though that detecting the same bioluminescent point source in shallower waters, where the background is brighter, would require eyes more similar to the medial eyes of *Phronima*. However, this has not been formally shown. It is clear from Land's statement that the selective pressure for detecting objects is likely strongest at the edges of a species' depth distribution and that dark objects become harder to detect at deeper depth and luminous objects at shallower depth.

If the primary function of *Phronima's* medial eyes is to detect dark objects against downwelling illumination, they would have to maintain a specific body orientation in the water column to ensure their medial eyes always point upwards. Behavioural observations *in situ* and in the laboratory [17], however, provided no evidence for this and Land [17] concluded that this hypothesis, at least in the case of *Phronima*, needs substantial revision and further investigation [17].

The model developed here provides an opportunity to directly compare the detection abilities of *Phronima*'s medial and lateral eyes for different visual tasks. We used a micro-computed tomography (micro-CT) approach [22] to extract optical parameters from two *Phronima sedentaria* specimens. We incorporated these parameters into our new model to directly compare the ability of *Phronima*'s medial and lateral eyes to detect different targets across different depths.

## Methods

### Detection distance of different targets in the midwater

Building on the computational models originally developed by Nilsson et al. [23] and extended by Nilsson et al. [9], we estimated the maximum detection distances of three type of targets; 1-bioluminescent point sources, 2-extended luminous objects, and 3-extended dark objects, under varying assumptions. Their modelling takes a statistical approach to visual detection in which the discrimination of a target against a background depends on the difference in photon counts between two channels. They assume there is a target channel, directed at the target, that detects an average of $N_T$ photons within a photoreceptive integration time, and a background channel, directed at the background (i.e., the light field adjacent to the target within the animal's visual field), that detects an average of $N_B$ photons per integration time. Photon counts are assumed to be sums of real photons and intrinsic noise ('false photons'). Photon counts are assumed to obey Poisson statistics [24,25], where the standard deviation of the photon catch is the square root of the mean ($N_T$ or $N_B$). Following Land [26] and Nilsson et al. [9], it is presumed that mean photon catches are high enough to assume Gaussian distributions for the photon counts about the means. Discrimination between the two channels is only possible when the difference between the two channels is greater than or equal to a reliability coefficient, $R$, times the standard deviation of the difference (which is the square root of the sum of the two means; [9,23,26]). To investigate the effects of spatial summation on visual performance, we assume each channel consists of a single ommatidium or a pool of summated neighbouring ommatidia from one eye only. The discrimination threshold from Eq 2.1 in Nilsson et al. [9] is given by:

$$|N_T - N_B| = R\sqrt{N_T + N_B} \tag{1}$$

We have adapted the equations from Nilsson et al. [9] for bioluminescent point sources and extended luminous objects to account for the properties of compound eyes. We also developed a new model for detection of semi-transparent objects. In addition, we examined the effect of spatial summation on the discrimination thresholds of all tasks. Definitions of all variables included in the equations are shown in Table 1. S1 Fig displays the radiance of light for the

**Table 1. Terms and values used in detection modelling.**

| Term | Value and Units | Definition |
|---|---|---|
| $N_T$ | photons | Mean number of real and false photons detected per integration time in a visual channel aimed at the target |
| $N_B$ | photons | Mean number of real and false photons detected per integration time in a visual channel viewing the background radiance |
| $R$ | 1.96 | Reliability coefficient, set for 95% confidence [26] |
| $N_{bio}$ | photons | Mean photon count originating from point sources |
| $N_{space}$ | photons | Mean photon count from background radiance |
| $X_{ch}$ | photons | Mean number of false photons (dark noise) per integration time in a visual channel |
| $q$ | 0.46 | Quantum capture efficiency [30] |
| $\Delta t$ | 0.037 s | Integration time [31] |
| $I_{space}$ | $6.28 \times 10^{15} \times 10^{-1.638 \times z}$ quanta m$^{-2}$ s$^{-1}$ sr$^{-1}$ | Downward radiance of space-light background at depth $z$ viewed at the position of the eye (see [23,27]) |
| | $5.11 \times 10^{13} \times 10^{-1.677 \times z}$ quanta m$^{-2}$ s$^{-1}$ sr$^{-1}$ | Horizontal radiance of space-light background at depth $z$ viewed at the position of the eye (see [23,27]) |
| $z$ | 200 m to 700 m in steps of 1 m | Depths from 200 m to 700 m. |
| $\kappa$ | 0.0385 m$^{-1}$ | Diffuse attenuation coefficient for downwelling background radiance (see [23,32]) |
| | 0 m$^{-1}$ | Diffuse attenuation coefficient for horizontal background radiance (see [23,32]) |
| $\alpha$ | 0.0468 m$^{-1}$ | Beam attenuation coefficient [23] |
| $d$ | See Table 2 | Rhabdom diameter (m) |
| $X$ | $6.7 \times 10^{-4}$ photons s$^{-1}$ | Dark-noise rate at 4˚C per ommatidium assuming five photoreceptors (~350 μm in length and ~10 μm in width) per ommatidium [30] |
| $T$ | 0.01 m | Object diameter (of targets other than point sources) assuming a circular object shape |
| $E$ | $10^{10}$ photons s$^{-1}$ (point source) $10^{9}$ photons s$^{-1}$ (each point source of the extended luminous object) | Number of photons emitted per second by a bioluminescent point source in all directions [33–35] |
| $A$ | See Table 2 | Corneal facet diameter of an ommatidium (m) |
| $r$ | See text | Distance between observer and target (m) |
| $f$ | See Table 2 | Focal length of a crystalline cone approximated by crystalline cone length (m) |
| $\Delta\phi$ | See Table 2 | Interommatidial angle (radians) |
| $\Delta\rho$ | See Table 2 | Acceptance angle of an ommatidium (radians), approximated using $d/f$ [36,37]. |
| $x$ | 0.001 m | Distance between photophores across the 1 cm diameter extended luminous object, meaning 49 point sources are exhibited across the object. |
| $n_{total}$ | See text | Number of summated ommatidia (see Section 1.2) |
| $S_i$ | See text | Relative sensitivity of i[th] neighbouring ommatidium to the point source |
| $S_{total}$ | See text | Summed relative sensitivity of the target channel to a point source (see Section 1.2) |
| $SP$ | See text | Relative sensitivity of an ommatidium to a single point source on an extended luminous object (see Section 2.1) |
| $S_l$ | See text | Summed relative sensitivity of an ommatidium to an extended luminous object (see Section 2.1) |
| $S_{ltotal}$ | See text | Summed relative sensitivity of the target channel to an extended luminous object (see Section 2.2) |
| $P$ | 49 | Number of point sources across the extended luminous object (see Section 2.1) |
| $N_{luminous}$ | photons | Number of photons received by the target channel (when it includes a single ommatidium) from an extended luminous object (see Section 2.1) |
| $N_{Sluminous}$ | photons | Number of photons received by the target channel (when it includes multiple ommatidia) from an extended luminous object (see Section 2.2) |
| $I_{object}$ | photons m$^{-2}$ s$^{-1}$ sr$^{-1}$ | Space-light radiance that enters the line of sight viewing an object with $F_T$ transparency factor (see Section 3) |
| $I_{opaue}$ | photons m$^{-2}$ s$^{-1}$ sr$^{-1}$ | Space-light radiance that enters the line of sight viewing an extended opaque object with $F_T = 0$ (see Section 3) |
| $F_T$ | See text | Relative transparency factor of extended dark objects which varies between 0 and 1, where 0 means the object is completely opaque (see Section 3). In our modelling the extended dark object has been set to 0.5 for 50% transparency. |
| $z_{object}$ | m | Depth at which the object is located (see Section 3) |

*(Continued)*

**Table 1.** (Continued)

| Term | Value and Units | Definition |
|---|---|---|
| $z_{observer}$ | m | Depth at which the observer is located (see Section 3) |
| $\theta$ | radians | Half the angle subtended by an extended dark object in visual space (see Section 3.1) |
| $\Omega_o$ | steradians | Solid angle of the target channel receptive field (comprising a single ommatidium) that is occupied by an extended dark object (see Section 3.1) |
| $\Omega_b$ | steradians | Solid angle of the target channel receptive field (comprising a single ommatidium) that is not occupied by an extended dark object (see Section 3.1) |
| $\Omega_F$ | steradians | Solid angle of a single ommatidium receptive field, equivalent to $1.133\Delta\rho^2$ (see S2 Appendix) |
| $\Omega_{osT}$ | steradians | Solid angle of the target channel receptive field (comprising multiple ommatidia) that is occupied by an extended dark object (see Section 3.2) |
| $\Omega_{bsT}$ | steradians | Solid angle of the target channel receptive field (comprising multiple ommatidia) that is not occupied by an extended object (see Section 3.2) |
| $N_{spaceT}$ | photons | Number of photons received by region of the receptive field that is not occupied by an extended dark object (see Section 3.1) |
| $N_{object}$ | photons | Number of photons received by region of the receptive field that is occupied by an extended dark object (see Section 3.1) |

different viewing directions and depths (radiance data originally from [27]; measured in the equatorial Pacific at 1005 hrs).

# 1. Discrimination of a point source

This case is applicable to the detection of a point source visualised against backgrounds of varying radiances. The target channel ($N_T$), whether derived from one or multiple ommatidia, is directed at the point source (Fig 2A and 2B). The target channel signal is compared to that of a background channel ($N_B$) having the same number of ommatidia aimed at the background next to the point source. From Nilsson et al. [9] it is assumed the target channel receives the same background space light ($N_{space}$) as the background channel and that both channels generate the same number of false photons ($X_{ch}$) per integration time.

*1.1 Discrimination of a point source by a single ommatidium.* Following Nilsson et al. [9], we assume a channel comprising a single ommatidium has a receptive field that is approximated by a two-dimensional Gaussian profile [28, 29]. We also assume that the point source is located at the centre of the receptive field where the relative sensitivity is equal to unity (Fig 2A and 2B, upper panels). Therefore, the target channel receives all photons from the point source ($N_{bio}$) that enter the channel within the channel's integration time. As shown in Nilsson et al. [9], the average photon count ($N_T$) of the target channel, comprising both real photons and false photons (noise), in this integration period, is:

$$N_T = N_{bio} + N_{space} + X_{ch}. \tag{1.1}$$

For the background channel, the average photon count ($N_B$) in the same integration time is:

$$N_B = N_{space} + X_{ch}. \tag{1.2}$$

By substituting Eqs 1.1 and 1.2 into Eq 1, the discrimination threshold (Eq 2.2 in [9]) is then:

$$\left|(N_{bio} + N_{space} + X_{ch}) - (N_{space} + X_{ch})\right| = R\sqrt{(N_{bio} + N_{space} + X_{ch}) + (N_{space} + X_{ch})} \tag{1.3}$$

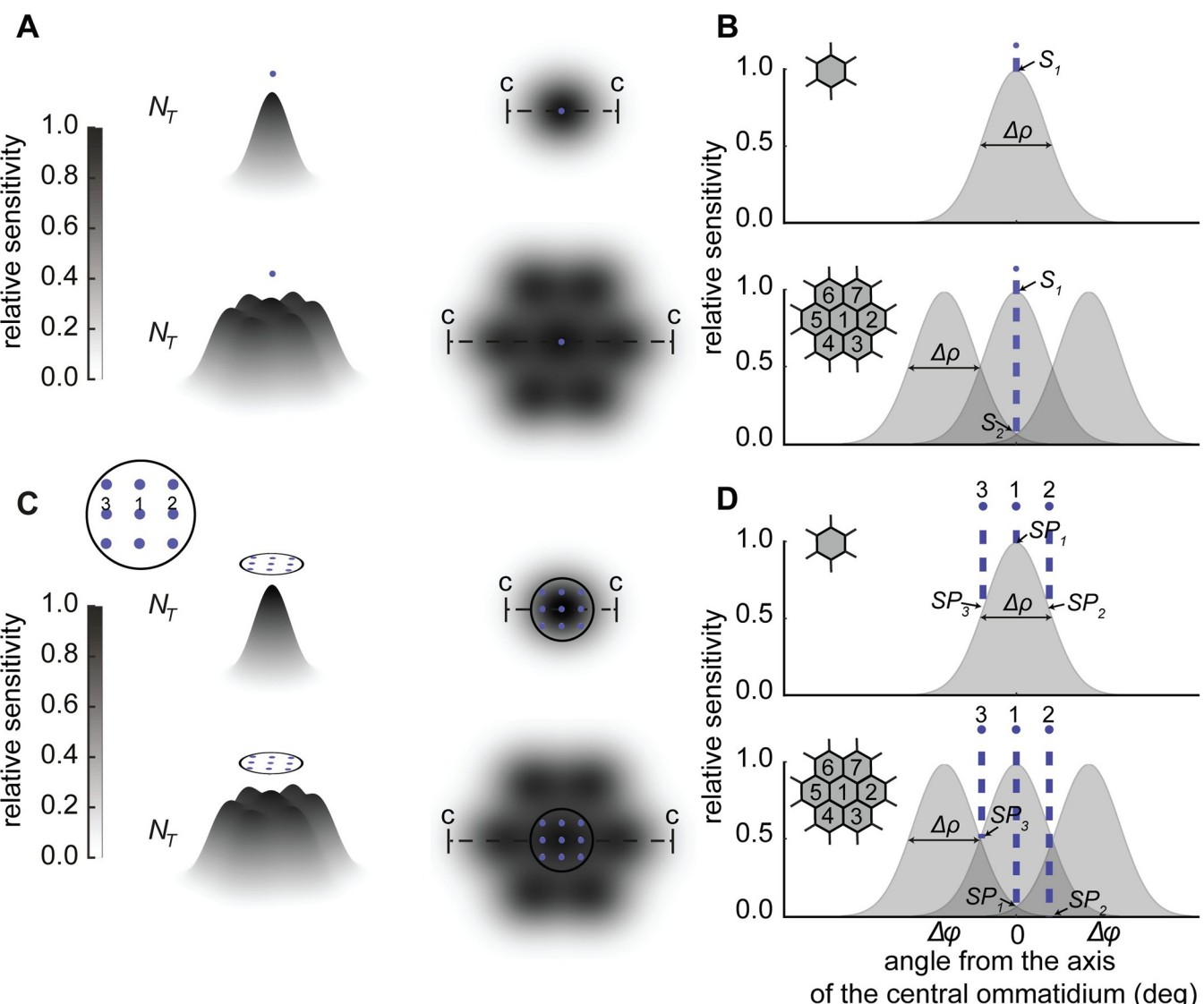

**Fig 2. The sensitivity of ommatidia to point sources.** (A) The receptive fields of target channels comprising either one ommatidium (upper panel) or seven ommatidia (lower panel) viewing a point source (small point depicted above receptive field). Grey levels represent the relative sensitivity of the receptive fields (darker indicating greater sensitivity) and the c-to-c dashed line represents the cross section through which the spatial sensitivity distributions shown in (B) relate. (B) Relative sensitivity of the ommatidia of target channels along the cross sections c-to-c in (A), showing the receptive fields of target channels comprising one ommatidium (upper panel) and seven ommatidia (lower panel). In the upper panel the relative sensitivity of the receptive field to the point source is denoted by $S_1$ and in the lower panel the relative sensitivity of ommatidium number two to the point source (see inset in the panel) is denoted by $S_2$. The hexagonal array inset in both panels shows the ommatidia that are included in the target channel. (C) The receptive fields of target channels comprising either one ommatidium (upper panel) or seven ommatidia (lower panel) viewing multiple point sources across a transparent object (the extended luminous object; multiple point sources represented by small points encased in black circle above receptive fields). The inset in (C) shows the point source labels as used in (D). (D) Relative sensitivity of ommatidia in the target channels along the cross sections c-to-c in (C), showing the receptive fields of target channels comprising one ommatidium (upper panel) and seven ommatidia (lower panel). In both panels of figure (D) the relative sensitivity of the receptive field to point sources one, two and three is denoted by $SP_1$, $SP_2$, and $SP_3$, but in the lower panel it is denoted for the receptive field of ommatidium five (see inset in the panel) only. The hexagonal array inset in both panels shows the ommatidia that are included in the target channel.

which simplifies to (Eq 2.3 from [9]):

$$N_{bio} = R\sqrt{N_{bio} + 2N_{space} + 2X_{ch}} \tag{1.4}$$

where $N_{bio}$, $N_{space}$ and $X_{ch}$ follow Eqs 2.4, 2.5, and 2.7, respectively from Nilsson et al. [9], as shown below.

$$N_{bio} = \frac{EA^2}{16r^2} e^{-\alpha \cdot r} q\Delta t \tag{1.5}$$

$$N_{space} = 1.133\Delta\rho^2 \left(\frac{\pi}{4}\right) A^2 q\Delta t \cdot I_{space} \tag{1.6}$$

$$X_{ch} = X\Delta t \tag{1.7}$$

All variables presented in these equations are shown in Table 1.

The background space light ($N_{space}$) depends on viewing direction and depth. The radiance of downwelling light when looking upwards is 200 times brighter than the radiance of upwelling light when looking downwards, and radiance decreases with increasing depth [9,27]. Naturally, light at depth will also vary with solar elevation (and hence latitude and time of day/year) and atmospheric weather conditions, none of which we explicitly consider here.

Ultimately, we require an equation that relates the maximum detection distance ($r$) to the eye parameters, the background radiance, and the brightness of the point source. The only variable that incorporates $r$ is $N_{bio}$ (Eq 1.5), therefore we have rearranged Eq 1.5 for $r$ (see S1 Appendix for full derivation) providing an equation for the maximum detection distance as follows:

$$r = \frac{2W\left(\frac{\alpha A \sqrt{Eq\Delta t}}{8\sqrt{N_{bio}}}\right)}{\alpha} \tag{1.8}$$

where $W$ is the Lambert W-function (the inverse function of $f(x) = xe^x$) and all other variables are shown in Table 1. To solve Eq 1.8, we have used the 'lambertw' function in MATLAB 2019b (The MathWorks Inc.), taking real rather than complex number output values. We now have an equation for $r$ in terms of $N_{bio}$. To find $N_{bio}$ we have rearranged Eq 1.4 for $N_{bio}$ in terms of $N_{space}$ and $X_{ch}$ as follows:

$$N_{bio} = 0.5(\pm R\sqrt{8N_{space} + R^2 + 8X_{ch}} + R^2) \tag{1.9}$$

and Eq 1.9 can now be substituted into Eq 1.8 to find $r$.

*1.2 Discrimination of a point source by a pool of spatially summated ommatidia.* In a channel comprising multiple ommatidia, we assume the total receptive field of the channel is the sum of several Gaussian profiles (Fig 2A and 2B, lower panels). Relative to the central ommatidium (ommatidium one in the inset of Fig 2B), neighbouring ommatidia (all other ommatidia in the inset of Fig 2B) are less sensitive to the point source and therefore fewer photons will enter the receptive field of these ommatidia. To calculate the relative sensitivity of a neighbouring ommatidium to the point source ($S_i$) we have evaluated the receptive field function of each neighbouring ommatidium at the position of the point source ($S_2$ in Fig 2B) in MATLAB 2019b (The MathWorks Inc.). The relative sensitivity can then be multiplied by $N_{bio}$ to give the number of photons received by each ommatidium in the target channel. The total photon catch of the target channel, $N_T$, is then:

$$N_T = \sum_{i=1}^{n_{total}} \left(N_{bio} \cdot S_i\right) + n_{total}N_{space} + n_{total}X_{ch} \tag{1.10}$$

which simplifies to:

$$N_T = N_{bio}S_{total} + n_{total}N_{space} + n_{total}X_{ch} \tag{1.11}$$

where $i$ is the $i$th ommatidium, $n_{total}$ is the total number of ommatidia in the target channel, and $S_{total}$ is the summed relative sensitivity to the point source of all ommatidia contributing to the target channel.

The background channel ($N_B$) will simply be:

$$N_B = n_{total}N_{space} + n_{total}X_{ch} \tag{1.12}$$

and substituting these into Eq 1 then gives:

$$|N_{bio}S_{total} + n_{total}N_{space} + n_{total}X_{ch} - (n_{total}N_{space} + n_{total}X_{ch})|$$
$$= R\sqrt{N_{bio}S_{total} + n_{total}N_{space} + n_{total}X_{ch} + (n_{total}N_{space} + n_{total}X_{ch})} \tag{1.13}$$

which simplifies to:

$$N_{bio}S_{total} = R\sqrt{N_{bio}S_{total} + 2n_{total}N_{space} + 2n_{total}X_{ch}}. \tag{1.14}$$

By rearranging Eq 1.14 for $N_{bio}$ we have:

$$N_{bio} = \frac{0.5(\pm R\sqrt{8n_{total}N_{space} + 8n_{total}X_{ch} + R^2} + R^2)}{S_{total}} \tag{1.15}$$

and to find $r$ we can substitute $N_{bio}$ as calculated with Eq 1.15 into Eq 1.8, above.

## 2. Discrimination of an extended luminous object

This case models the visibility of a transparent animal, such as a ctenophore, that exhibits multiple bioluminescent point sources across its body (an extended luminous object). Once again, the target channel ($N_T$) is directed at the extended luminous object, and its signal is compared to that of a background channel ($N_B$) aimed at the background next to the object. Because the extended luminous object is transparent both channels receive the same number of photons from the background space light ($N_{space}$) and, as both channels have the same number of photoreceptors, we assume that they both generate the same number of false photons ($X_{ch}$) in each integration time. The target channel also receives photons from the combined output of multiple point sources each of which we have set to emit $10^9$ photons per second over $4\pi$ steradians providing a comparable saliency to the single point source. We assume that point sources are packed in a square array across the object and that the object is centred on the receptive field (Fig 2C).

*2.1 Discrimination of an extended luminous object by a single ommatidium.* Because we assume the sensitivity of the receptive field is approximated by a two-dimensional Gaussian function, the relative sensitivity to a point source viewed off centre will be less than the sensitivity to a point source viewed at the centre. To account for this, we have evaluated the receptive field function at the position of each point source to find each respective relative sensitivity ($SP_j$), where $j$ is the $j$th point source. The sum of the relative sensitivities to all point sources ($S_l$) multiplied by $N_{bio}$ provides the total number of photons received from the extended luminous object, which we have termed $N_{luminous}$. Therefore, $N_{luminous}$ will be:

$$N_{luminous} = N_{bio} \times \sum_{j=1}^{P} SP_j = N_{bio}S_l \tag{2.1}$$

where $P$ is the number of point sources, $j$ is the $j$th point source, and $S_l$ is the relative sensitivity to the extended luminous object of the target channel (which we have numerically calculated). The photon catch (in the modelled integration time) of the target channel is then given by Eq 1.1, but with $N_{luminous}$, substituted for $N_{bio}$ and the background channel is given by Eq 1.2. The discrimination threshold then becomes:

$$N_{luminous} = R\sqrt{N_{luminous} + 2N_{space} + 2X_{ch}}. \tag{2.2}$$

Substituting $N_{luminous}$ for $N_{bio}$ in Eq 1.8 we can calculate r as:

$$r = \frac{2W\left(\frac{\alpha A \sqrt{Eq\Delta t S_l}}{8\sqrt{N_{luminous}}}\right)}{\alpha} \tag{2.3}$$

*2.2 Discrimination of an extended luminous object by a pool of spatially summated ommatidia.* If the target channel now consists of $n_{total}$ spatially summated ommatidia (Fig 2C and 2D, lower panels), the relative senstivity of each ommatidium to each point source ($SP_j$) will need to be evaluated separately using Eq 2.1. The number of photons from the extended luminous object that are received by the target channel is then the sum of $N_{luminous}$ from all ommatidia within the target channel which we have termed $N_{Sluminous}$:.

$$N_{Sluminous} = \sum_{i=1}^{n_{total}} N_{luminous_i} = N_{bio} \times \sum_{i=1}^{ntotal} S_{li} = N_{bio} S_{ltotal} \tag{2.4}$$

where $N_{luminous_i}$ is the number of photons recieved by the $i$th ommatidium within the target channel, $S_{li}$ is the total senstivity of the $i$th ommatidium (calculated numerically), and $S_{ltotal}$ is the relative sensitivity of the target channel to the extended luminous object. The target channel then becomes:

$$N_T = N_{Sluminous} + n_{total}N_{space} + n_{total}X_{ch}. \tag{2.5}$$

The background channel photon count, $N_B$, will be the same as Eq 1.12.
The discrimination threshold following Eq 1 is then:

$$(N_{Sluminous} + n_{total}N_{space} + n_{total}X_{ch}) - (n_{total}N_{space} + n_{total}X_{ch})$$
$$= R\sqrt{(N_{Sluminous} + n_{total}N_{space} + n_{total}X_{ch}) + (n_{total}N_{space} + n_{total}X_{ch})} \tag{2.6}$$

which simplifies to:

$$N_{Sluminous} = R\sqrt{N_{Sluminous} + 2n_{total}N_{space} + 2n_{total}X_{ch}}. \tag{2.7}$$

By substituting $S_l$ with $S_{ltotal}$ and $N_{luminous}$ with $N_{Sluminous}$ in Eq 2.3 we can then find the maximum detection distance $r$, in a manner analogous to the cases above, $r$ being given by:

$$r = \frac{2W\left(\frac{\alpha A \sqrt{Eq\Delta t S_{ltotal}}}{8\sqrt{N_{Sluminous}}}\right)}{\alpha}. \tag{2.8}$$

# 3. Discrimination of extended dark objects

This case models the visibility of extended dark objects against varying background radiances. The extended dark object may completely block the background radiance, or it may be semi-

transparent. We modified the radiance model of Preisendorfer [38] to calculate the space-light that enters the line of sight as follows:

$$I_{object} = F_T I_{space}(z_{object})e^{(-\alpha \cdot r)} + I_{space}(z_{observer})(1 - e^{(\kappa - \alpha)r}) \qquad (3.1)$$

where $I_{object}$ is the apparent radiance of the object as seen by the observer, $I_{space}(z_{object})$ and $I_{space}(z_{observer})$ are space-light at the depth of object and observer, respectively. $F_T$ is the transparency factor which ranges between 0 and 1. For an extended opaque object $F_T$ is 0 so Eq 3.1 reduces to:

$$I_{opaque} = I_{space}(z_{observer})(1 - e^{(\kappa - \alpha)r}) \qquad (3.2)$$

which is the extended opaque object radiance model used by Nilsson et al. [7].

*3.1 Discrimination of extended dark objects by a single ommatidium.* Unlike Nilsson et al. [7], we assume the receptive field of a single ommatidium has a two-dimensional Gaussian sensitivity profile and that the visualised object can be smaller than the receptive field (Fig 3). Therefore, the object occupies an $\Omega_o$ solid angle of the receptive field. The number of photons received by the region of the receptive field that is occupied by the object is termed $N_{object}$:

$$N_{object} = \Omega_o \left(\frac{\pi}{4}\right) A^2 q \Delta t \cdot I_{object} \qquad (3.3)$$

where Eq 3.3 incorporates the sensitivity of the eye (using the solid angle of the receptive field that is occupied by the object, $\Omega_o$; the area of the ommatidial facet, $\left(\frac{\pi}{4}\right)A^2$; the quantum efficiency of the photoreceptors, $q$; and the integration time of the photoreceptors, $\Delta t$) and the amount of light available to the eye ($I_{object}$). We assume the centre of the object is at the centre of the receptive field of the target channel. The receptive field of the target channel (assuming a Gaussian function) has a solid angle of $1.133\Delta\rho^2$ steradians and the object occupies some proportion of that solid angle set by $2\theta$, where $\theta$ is half the angle (in radians) of the object in visual space. The solid angle of the receptive field that is occupied by the object is therefore:

$$\Omega_o = 1.133(\Delta\rho)^2 \left(1 - e^{-4\ln 2 \times \left(\frac{\theta}{(\Delta\rho)}\right)^2}\right). \qquad (3.4)$$

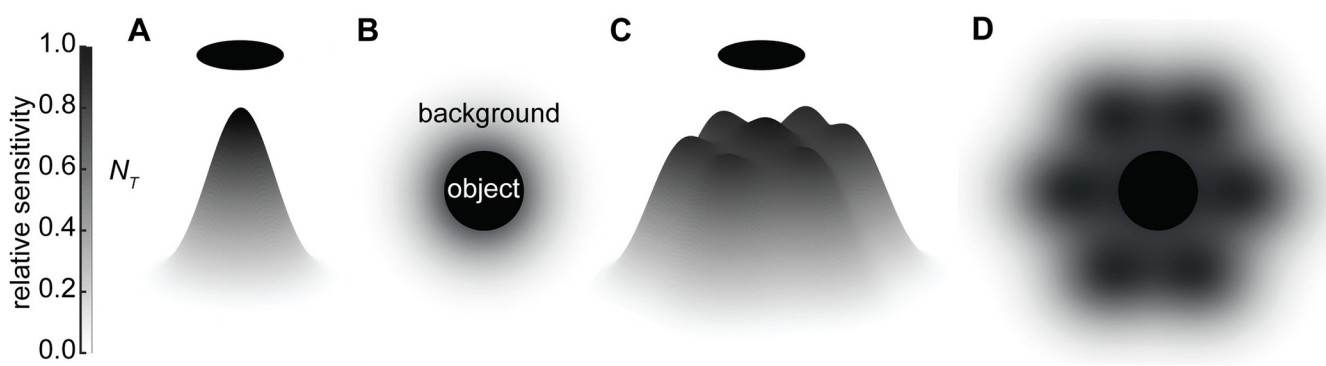

**Fig 3.** The receptive fields of target channels comprising (A) a single ommatidium or (C) seven summed ommatidia viewing an extended dark object against background radiance. (B, D) Top-down views of the receptive fields in (A) and (C) showing the object obscuring parts of the receptive field (black circle) and the remaining parts of the receptive field that view the background. Grey levels represent the relative sensitivity of the receptive fields (darker indicating greater sensitivity).

Full derivations of Eq 3.4 can be found in the S2 Appendix. The remaining light that could be received by the target channel comes from background radiance and enters the eye through the part of the receptive field that is not covered by the object ($\Omega_b$). Therefore:

$$\Omega_b = 1.133(\Delta\rho)^2 e^{-4\ln 2 \times \left(\frac{\theta}{(\Delta\rho)}\right)^2}. \tag{3.5}$$

For full derivations of $\Omega_b$ see the S2 Appendix.

We have termed the remaining background radiance that can be received by the target channel as $N_{spaceT}$ which can be calculated as:

$$N_{spaceT} = \Omega_b \left(\frac{\pi}{4}\right) A^2 q \Delta t \cdot I_{space} \tag{3.6}$$

which can be rearranged to:

$$N_{spaceT} = \frac{\Omega_b}{1.133\Delta\rho^2} N_{space}. \tag{3.7}$$

The target channel is therefore:

$$N_T = N_{object} + N_{spaceT} + X_{ch} \tag{3.8}$$

and the background channel remains the same as that presented in Eq 1.2. The discrimination threshold from Eq 1 then becomes:

$$|(N_{object} + N_{spaceT} + X_{ch}) - (N_{space} + X_{ch})| = R\sqrt{(N_{object} + N_{spaceT} + X_{ch}) + (N_{space} + X_{ch})} \tag{3.9}$$

which simplifies to:

$$|N_{object} + N_{spaceT} - N_{space}| = R\sqrt{N_{object} + N_{spaceT} + N_{space} + 2X_{ch}}. \tag{3.10}$$

By substituting the equation for $N_{spaceT}$ presented in Eq 3.7, Eq 3.10 reduces to:

$$\left|N_{object} + \left(\frac{\Omega_b}{1.133\Delta\rho^2} - 1\right)N_{space}\right| = R\sqrt{N_{object} + \left(1 + \frac{\Omega_b}{1.133\Delta\rho^2}\right)N_{space} + 2X_{ch}}. \tag{3.11}$$

If the object and the observer are aligned horizontally at the same depth, so $\kappa = 0$ (Table 1), we can rearrange Eq 3.3 for $r$ as follows:

$$r = \frac{1}{-\alpha}\ln\frac{\Omega_o\left(\frac{\pi}{4}\right)A^2 q \Delta t \cdot I_{space} - N_{object}}{(1 - F_T)\Omega_o\left(\frac{\pi}{4}\right)A^2 q \Delta t \cdot I_{space}} \tag{3.12}$$

and $N_{object}$ can be derived from Eq 3.11 as below and substituted into Eq 3.12 to find $r$.

$$N_{object} = \pm 0.5\left(R\sqrt{8N_{space} + R^2 + 8X_{ch}} + R^2\right) + \left(1 - \frac{\Omega_b}{1.133\Delta\rho^2}\right)N_{space} + \frac{R^2}{2} \tag{3.13}$$

For a case where the object and observer are at different depths we numerically solved for the maximum detection distance.

*3.2 Discrimination of extended dark objects by a pool of spatially summated ommatidia.* In this case each channel comprises $n_{total}$ ommatidia and for each ommatidium in the target channel we have numerically calculated the solid angle of the receptive field that views the object ($\Omega_{oi}$) and the solid angle of the receptive field that views the background radiance ($\Omega_{bi}$) as above. For simplicity, we define $\sum_{i=1}^{n_{total}} \Omega_{oi}$ as the summated solid angle of the target channel

that views the object ($\Omega_{osT}$), and $\sum_{i=1}^{n_{total}} \Omega_{bi}$ as the summed solid angle of the target channel that views the background radiance ($\Omega_{bsT}$), where $i$ is the $i$th ommatidium. Therefore, $N_{object}$ will be:

$$N_{object} = \Omega_{osT}\left(\frac{\pi}{4}\right)A^2 q\Delta t \cdot I_{object} \qquad (3.14)$$

where $I_{object}$ is calculated from Eq 3.1. $N_{spaceT}$ will be:

$$N_{spaceT} = \Omega_{bsT}\left(\frac{\pi}{4}\right)A^2 q\Delta t \cdot I_{space} \qquad (3.15)$$

which can be simplified to:

$$N_{spaceT} = \frac{\Omega_{bsT}}{1.133\Delta\rho^2} \cdot N_{space}. \qquad (3.16)$$

The target channel is then given by:

$$N_T = N_{object} + N_{spaceT} + n_{total}X_{ch} \qquad (3.17)$$

and the background channel is:

$$N_B = n_{total}N_{space} + n_{total}X_{ch}. \qquad (3.18)$$

The discrimination threshold from Eq 1 then becomes:

$$|(N_{object} + N_{spaceT} + n_{total}X_{ch}) - (n_{total}N_{space} + n_{total}X_{ch})|$$
$$= R\sqrt{(N_{object} + N_{spaceT} + n_{total}X_{ch}) + (n_{total}N_{space} + n_{total}X_{ch})}\,(3.19)$$

which reduces to:

$$|N_{object} + (\frac{\Omega_{bsT}}{1.133\Delta\rho^2} - n_{total})N_{space}| = R\sqrt{N_{object} + (\frac{\Omega_{bsT}}{1.133\Delta\rho^2} + n_{total})N_{space} + 2n_{total}X_{ch}}. \quad (3.20)$$

If the object and the observer are aligned horizontally at the same depth, so $\kappa = 0$ (Table 1), we can rearrange Eq 3.14 for $r$ as follows:

$$r = \frac{1}{-\alpha}\ln\frac{\Omega_{osT}\left(\frac{\pi}{4}\right)A^2 q\Delta t \cdot I_{space} - N_{object}}{(1 - F_T)\Omega_{osT}\left(\frac{\pi}{4}\right)A^2 q\Delta t \cdot I_{space}} \qquad (3.21)$$

and then solve Eq 3.20 for $N_{objectT}$ giving:

$$N_{object} = \pm 0.5\left(R\sqrt{8n_{total}N_{space} + 8n_{total}X_{ch} + R^2}\right) + \left(n_{total} - \frac{\Omega_{bsT}}{1.133\Delta\rho^2}\right)N_{space} + \frac{R^2}{2} \quad (3.22)$$

which can be substituted into Eq 3.21 to find $r$. We again solved numerically for the maximum detection distance for a case where the object and the observer are at different depths.

## Optical measurements of *Phronima sedentaria*

Two specimens of *P. sedentaria* were collected in July 2018 on board the R/V *Hugh R. Sharp* (University of Delaware) using a midwater Tucker trawl (1.5 m x 1.5 m opening, 500 μm mesh) deployed off Lewes, Delaware (37˚42'15"N, 73˚37'21.8"W) at a depth of approximately 600 m. Specimens were identified using the identification keys provided by Zeidler [39] and hereinafter are referred to as *Phronima*. The heads of the specimens were dissected from the

body and then fixed in 2% cacodylate buffered glutaraldehyde. The heads were then stained in a 70% ethanol/0.5% phosphotungstic acid (PTA) solution for 30 days before 3D micro-CT X-ray scanning. Before scanning, specimens were mounted in 500 µl sealed Eppendorf tubes containing 0.5% low temperature gelling agarose solution. Specimens were scanned with the GE Phoenix v|tome|x M 180 kV Nano Tube micro-CT at the Smithsonian National Museum of Natural History. Scans were conducted at a voltage of 90 kV and 4.26 W with 36.6x optical magnification to deliver an isotropic voxel size of 5.46 µm for *Specimen* 1 and 5.69 µm for *Specimen* 2. Raw projection data were reconstructed using datos|X (GE Sensing and Inspection Technologies GmbH) and visualized and exported using VG Studio (Volume Graphics).

The analyses of the reconstructed micro-CT images followed that of Bagheri et al. [22]. Briefly, 3D optical axes of each ommatidium of the compound eyes were calculated by manually marking the centre of the corneal facet and the centre of the proximal tip of the crystalline cone (Fig 4) using the custom-made software described in Bagheri et al. [22] developed in MATLAB 2019b (The MathWorks Inc.). The software allowed us to match these corneal and crystalline cone points by aligning the selected ommatidia within three perpendicular planes (i.e., xy-, yz-, xz-planes). Each ommatidium was labelled with horizontal row and vertical column numbers which were used to identify the neighbouring ommatidia in subsequent processing of interommatidial angles. The raw data were smoothed using an averaging algorithm

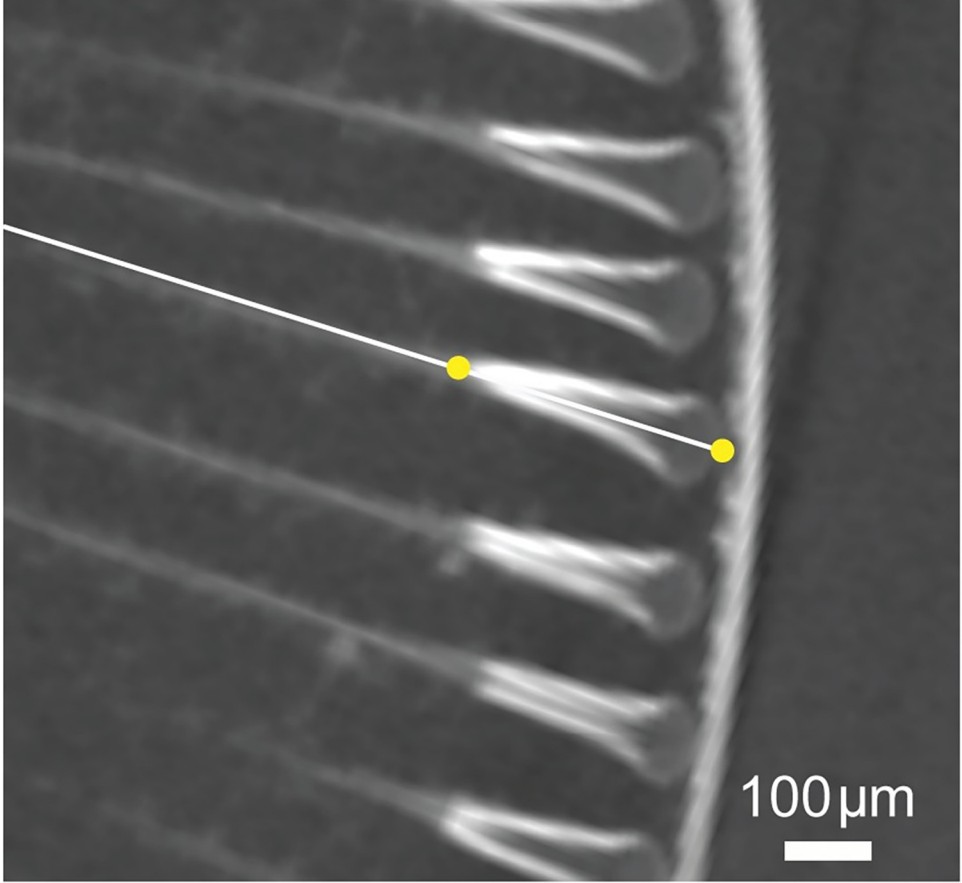

**Fig 4. Micro-CT section image of the medial eye of *P. sedentaria* showing the distal crystalline cone abutting the long light guide proximally.** The two marked points, one central on the cornea and the other at the junction between the crystalline cone and the light guide, determine the optical axis of the ommatidium.

in which the viewing direction of each ommatidium was averaged from the viewing directions of its six neighbours over five iterations. Interommatidial angles were calculated by computing the angles between the optical axes of the select ommatidium and neighbouring ommatidia. The average distances between the corneal points of each ommatidium and its six adjacent neighbouring ommatidia were used to estimate the facet diameters of the ommatidia.

In the medial eyes of *Phronima* the distal portion of the crystalline cone advances into a long light guide portion of the crystalline cone that delivers light to the rhabdom [10]. Therefore, the only light focusing that occurs is between the cornea and the junction between the distal crystalline cone and the light guide portion of the crystalline cone. The length of the crystalline cones were estimated by marking a point on this junction and the cornea (Fig 4), and these lengths approximated the focal lengths of the ommatidia, a reasonable approximation, as shown by Land [10]. Rhabdom diameters were estimated by visualizing the micro-CT images and using the measurement tool in Dragonfly software (Object Research Systems Inc.). The diameters of 50 rhabdoms were measured in both medial and lateral eyes yielding average rhabdom diameters for each eye sub-type. Extracted visual parameters were used as inputs for the computational models. To test the sensitivity of the models to different visual parameters we varied the parameters by a factor of two.

## Results

The eyes of both *Phronima* specimens were similar in size and structure. The medial eyes (blue shaded areas in Fig 1D), measured dorsoventrally, were 6.3 and 6.2 mm long and contained 424 and 428 ommatidia (specimen 1 and 2 respectively) while the lateral eyes (red shaded areas in Fig 1D) contained 261 and 265 ommatidia in specimens 1 and 2, respectively. Facet diameters were larger while interommatidial angles and acceptance angles were smaller in the medial eyes compared to the lateral eyes (Table 2).

### Effect of optical parameters on detection distance

Fig 5 shows the detection distances predicted by the model at various facet diameters (*A;* Fig 5A–5C), quantum efficiencies/integration time (*q/t*; Fig 5D–5F), and acceptance angles (*Δρ*; Fig 5G–5I) for different targets. Not surprisingly, the model predicted an inverse relationship

**Table 2. Average medial and lateral right eye parameters for two *Phronima* specimens based on measurements from micro-CT reconstructions.** Measurements are given as mean ± standard deviation. For each eye type, the average and standard deviation for facet diameter and rhabdom width were calculated from all ommatidia within the right eye of each specimen (N = 424 and 428 for the medial eyes and N = 261 and 265 for the lateral eyes), using these values, we then calculated the average between the two specimens (for both means and standard deviations). The average and standard deviation for focal length were calculated with a similar method but the measurements were taken from a subset of ommatidia within the right eye of each specimen (N = 50 for each eye type). Acceptance angles in degrees approximated using $d/f$ using the average values of $d$ and $f$ (converted from radians; [36,37]). "No. ommatidia summated" represents the number of spatially summated neighbouring ommatidia.

| Parameter | Medial eye ($N_{eyes}$ = 2) | Lateral eye ($N_{eyes}$ = 2) |
|---|---|---|
| Facet diameter (*A*, μm) | 156.7±8.2 | 112.9±24.5 |
| Rhabdom width (*d*, μm) | 22.3±3.2 | 43.3±8.5 |
| Focal length (*f*, μm) | 330.1±36.5 | 242.7±40.1 |
| Acceptance angle (*Δρ*, deg) | 3.9 | 10.2 |
| Interommatidial angle (*Δφ*, deg) | 1.6±1.2 | 10.5±2.2 |
| *Δρ:Δφ* ratio | 2.4 | 1.0 |
| No. ommatidia summated | 19 | 1 |

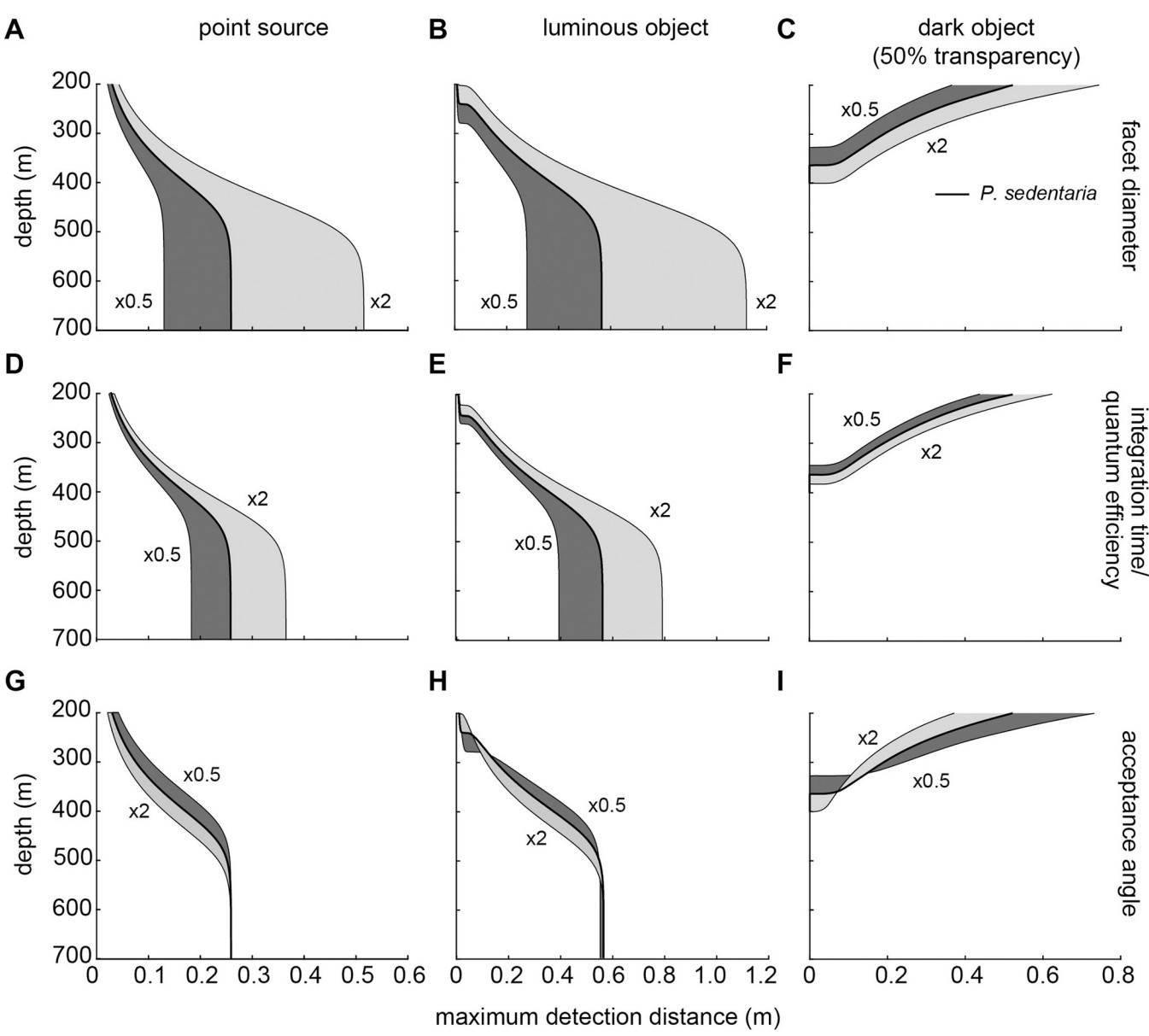

**Fig 5. The effects of anatomical parameters on detection distances of three targets against downwelling radiance modelled across depths.** Rows show the effects of increasing or decreasing the test parameter by a factor of two on the maximum detection distance. (A–C) facet diameter, (D–F) quantum efficiency or integration time (exactly equivalent effects), and (G–I) acceptance angle of the ommatidium. Columns show the model results for (A, D, G) a point source, (B, E, H) an extended luminous object, and (C, F, I) an extended dark object with 50% transparency. Extended objects were 1 cm in diameter. Results for an extended dark object with 0% transparency are given in S2 Fig. The thickest black line in each figure shows the result from the average medial eye parameters based on our *Phronima* specimens (Table 2). The thinner black lines bounding the shaded areas show the result of decreasing (dark grey) or increasing (light grey) the parameters by a factor of two. Note the different scale of the x-axis for different targets.

between detectability of all targets with depth. Dark object detection was best in shallow (more well-lit) versus deep (less well-lit) ocean depths, and detectability of both point sources and luminous objects was easier at greater depths where there is greater contrast with the background.

Facet diameter had a strong effect on the maximum detection distances for all targets. Increasing the facet diameter by a factor of two led to an improvement at all depths for all

targets but this improvement was smaller for dark objects compared to luminous objects or point sources (Fig 5A–5C). For dark objects detection distances improved most significantly at shallower depths with a maximum improvement of 42% at 200 m (Figs 5C and S2). In contrast, larger facets increased detection distances for both luminous objects and point sources most strongly at deeper depths (below ~550 m), with a maximum improvement of 98% (Fig 5A and 5B).

Increased quantum efficiency and integration time had the exact same effect on maximum detection distance for all targets thus the results were combined (Fig 5D–5F). When quantum efficiency or photoreceptor integration time was increased, detection distances increased for both point sources and luminous objects to a maximum of 41% at 550 m, and 19% for dark objects at 200 m (Figs 5D–5F and S2). The improvement in maximum detection distance from increased facet diameter was greater (Fig 5A–5C) than the improvement from increased quantum efficiency / photoreceptor integration time (Fig 5D–5F).

In contrast to increased facet diameter and integration time/quantum efficiency above, increased photoreceptor acceptance angle decreased detection distance for all targets at most, but not all, depths (Fig 5G–5I). For example, doubling the acceptance angle decreased the detection distances of a point source at 200 m depth by 42% (Fig 5G). However, at depth limits with the least contrast (shallow for luminous objects and deeper for dark objects) increasing acceptance angle increased detection distance for both extended objects. This inversion took place at approximately 250 m for luminous and 360 m for dark objects. Effects of each parameter on the detection distance of a dark object with 0% transparency were approximately the same as for a dark object with 50% transparency and are shown in the S2 Fig. Below 570 m, the acceptance angle no longer had any effect on the detection of point sources or luminous objects (Fig 5G and 5H).

## Spatial summation can improve maximum detection distances

To explore the effects of spatial summation on visual performance, we compared the maximum detection distances of two different eye 'designs' (Fig 6). In the first eye design ommatidial acceptance angles ($\Delta\rho$), were set as equal to the interommatidial angles ($\Delta\phi$; Fig 6A–6C), which is typical for imaging systems optimised for spatial resolution [40] and were found in *Phronima* lateral eyes (Table 2). We called this design 'optimal sampling'. In the second eye design, acceptance angles were 2.4 times greater than the interommatidial angles (Fig 6D–6F), as found in *Phronima's* medial eyes (Table 2). We called this design 'oversampling'. We also compared the effects of different extents of spatial summation on detection distance by including a single ommatidium, seven ommatidia, or 19 ommatidia in the summated pool of the target channel.

In an eye that has optimal sampling (e.g. *Phronima*'s lateral eye), the effect of spatial summation on the maximum detection distance of each target differed depending on the type of target. At depths below 500 m, spatial summation always improved detection distance for both point sources and luminous objects. Detection of point sources showed the largest improvements. At approximately 450 m detection distance more than doubled for 7 and 19 ommatidia. At shallower depths, the relative improvement in detection distance of point sources was smaller, with a 33% improvement at 300 m. Spatial summation only improved detection distance of the extended luminous object at depths shallower than 254 m and deeper than 485 m (Fig 6B). Moreover, even at depths where spatial summation increased the detection distance of the extended luminous object, the relative increase in distance was small, less than 20% at maximum. Spatial summation allows detection of dark objects at greater depths (~370 m), but above 370 m it decreased detection distance (Fig 6C).

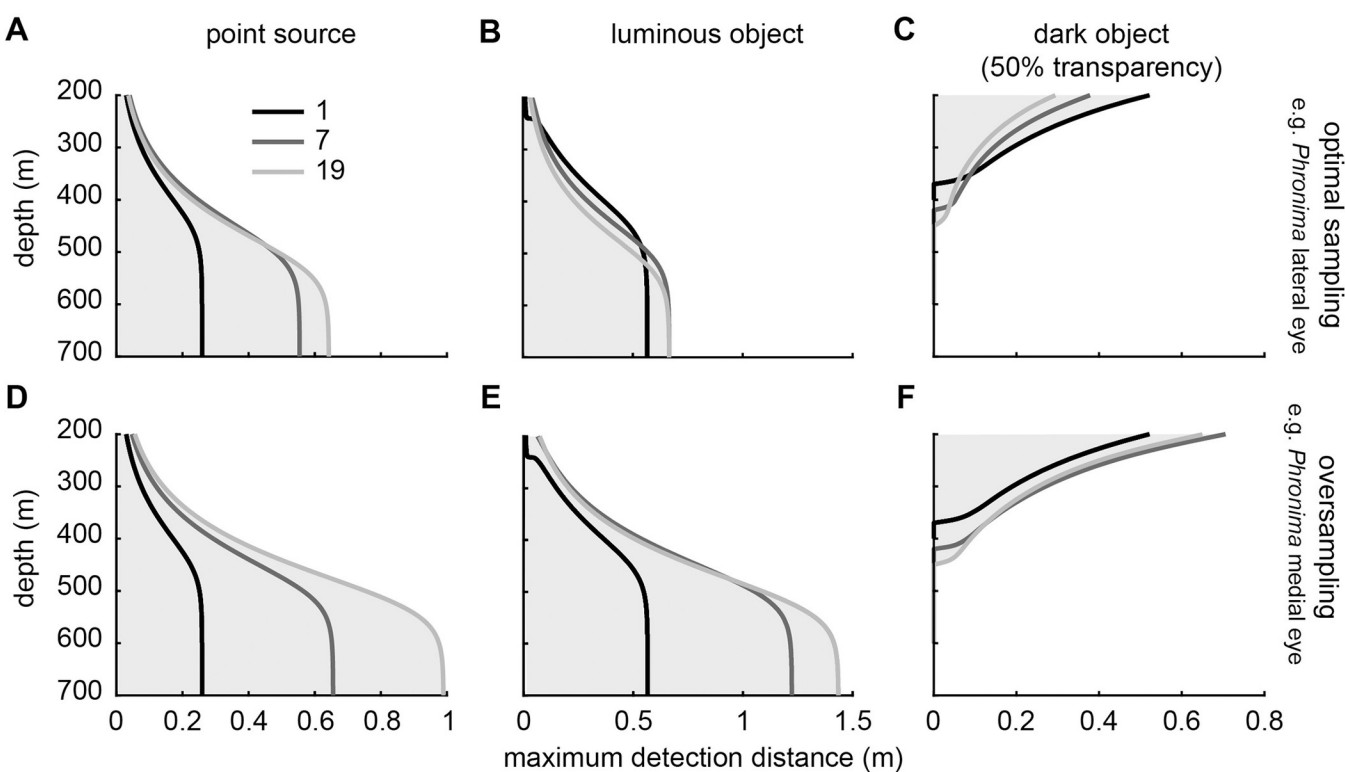

**Fig 6. Visual detection distance across depths showing the effect of spatially summating different numbers of ommatidia in eyes with different sampling arrangements.** Rows show the maximum detection distances of eyes that employ spatial summation and either (A–C) optimally sample visual space ($\Delta\rho{:}\Delta\phi$ = 1) or (D–F) oversample visual space ($\Delta\rho{:}\Delta\phi$ = 2.4). Each column shows the model results for (A, D) a point source, (B, E) an extended luminous object, and (C, F) an extended dark object. Extended objects were 1 cm in diameter. All targets were viewed against downwelling radiance at different depths. In all figures, the model results are shown for a single ommatidium (black), seven spatially summated ommatidia (dark grey), and 19 spatially summated ommatidia (light grey). Grey shading shows the distances and depths at which detection can occur. Model inputs were those of the medial eyes of *Phronima* (Table 2) and model parameters (Table 1).

Spatial summation in an eye that oversamples (e.g. *Phronima*'s medial eye) improves all detection distances for all target types. However, the amount of improvement in detection distance differed depending on target type, the number of ommatidia being spatially summated, and depth. The maximum improvement in detection distance of 19 summated ommatidia, versus a single ommatidium, is 280% for the point source, 154% for the extended luminous object (below ~570 m; Fig 6D and 6E), and 52% for the extended dark object (below 370 m; Fig 6F). Similar to the optimal sampling eye, spatial summation also increased the depth at which the extended dark object was detectable (from 370 to 450 m; Fig 6F). Increasing the extent of spatial summation, or the number of ommatidia being spatially summated, always improved detection distances for the point source (Fig 6D). However, increasing the extent of spatial summation only improved the detection distances for the extended luminous and dark objects at deeper depths. For example, spatially summating 19 ommatidia compared with seven, only improved detection distances at depths below 475 m for the extended luminous object and below 400 m for an extended dark object, though spatial summation of any kind always outperform an individual ommatidium (Fig 6D–6F). For *Phronima* to achieve the same improvement in visual range using temporal summation instead of spatial summation, the integration time would need to increase from 37 ms to 550 ms for a point source, to 518 ms for an extended luminous object, and to 83 ms for an extended dark object.

## The effect of ommatidial overlap on detection distances

Increasing the overlap of neighbouring ommatidia by decreasing the interommatidial angle, increases detection distances (Fig 7A–7C) but inevitably decreases the overall receptive field size of the target channel and the animal's overall field of view. This leads to a clear trade-off between detection distance and field of view (Fig 7D–7F). This trade-off could be expected to lead to an optimum in the volume of water the animal can survey at any point in time since search volume is a function of both detection distance and visual field size. However, there is no optimum in the eye's search volume as a function of ommatidial overlap (S3 Fig).

## *Phronima*'s medial vs lateral eyes

In Fig 7 (dotted lines), we used our measured ratio of $\Delta\rho{:}\Delta\phi$ to estimate the number of spatially summated ommatidia based on Land's [10] suggestions (7 ommatidia). However, since summating 19 ommatidia slightly improves detection distances (Fig 6), we modelled the functional

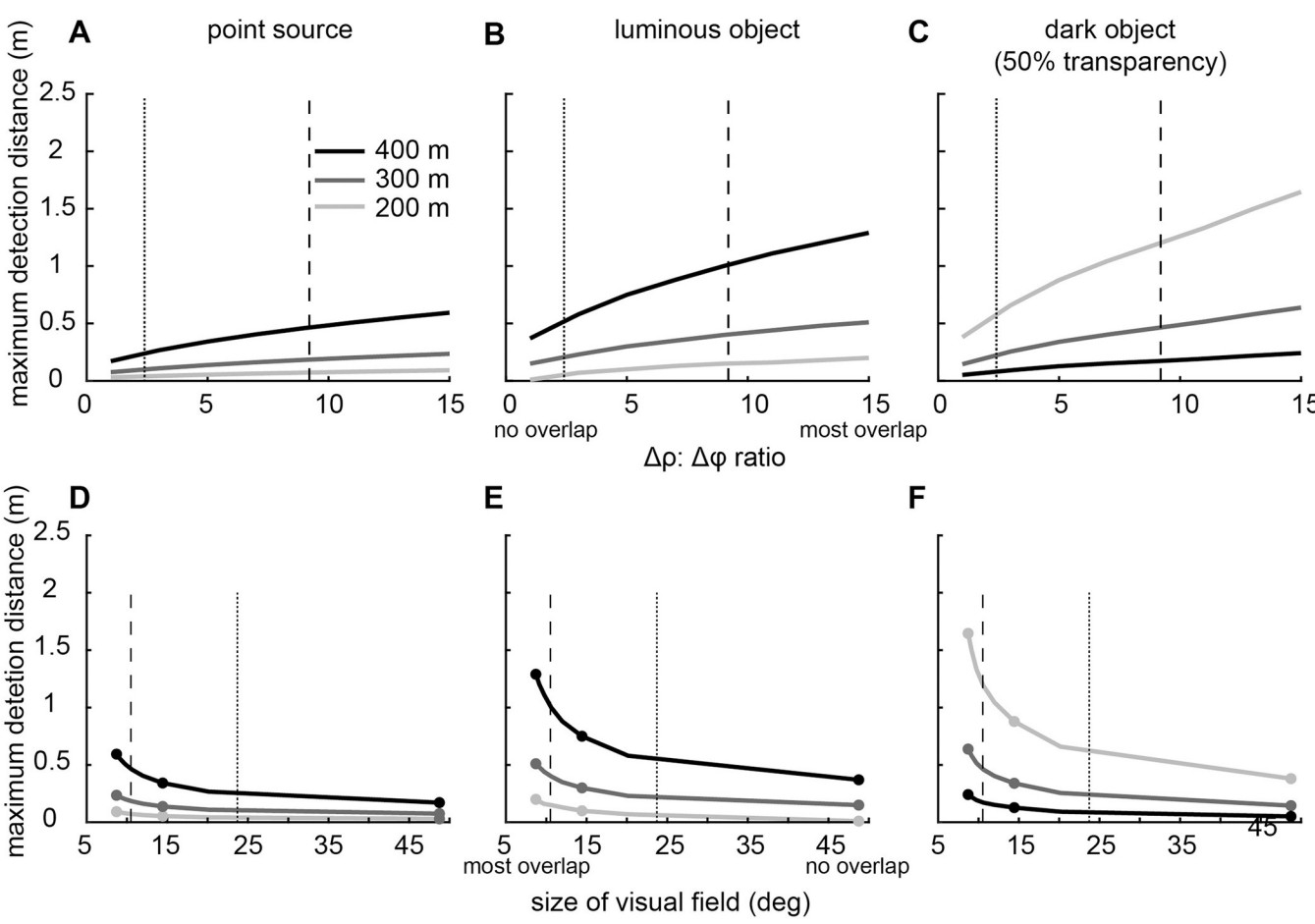

**Fig 7. Effect of overlap of ommatidial receptive fields on detection distances and the size of visual field.** (A-C) Increasing overlap always improved maximum detection distance for all targets. (D-E) However, there is a clear trade-off between detection distance and the size of field of view due to ommatidial overlap. Columns show the model results for the detection of a point source (A, D), an extended luminous object (B, E), and an extended dark object (C, F). The number of ommatidia summating in a single channel was calculated as a function of $\Delta\rho{:}\Delta\phi$ ratio [10]. For overlaps greater than 16, there are not enough ommatidia within an eye of *Phronima* to from two distinct channel. The size of the field of view and the receptive field size of the target channel were calculated using the derivations shown in the S3 Appendix. The acceptance angle was taken from the medial eyes of *Phronima*, 3.9˚ (Table 2), and we varied the interommatidial angle (from 0.26 to 3.9˚) to model different $\Delta\rho{:}\Delta\phi$ ratios. Dotted vertical lines show the overlap measured in our *Phronima* specimens ($\Delta\rho{:}\Delta\phi$ = 2.4) and dashed vertical lines show the overlap measured by Land [6] ($\Delta\rho{:}\Delta\phi$ = 9.2). Small circles in (D-F) shows $\Delta\rho{:}\Delta\phi$ ratios of 1, 5 and 15.

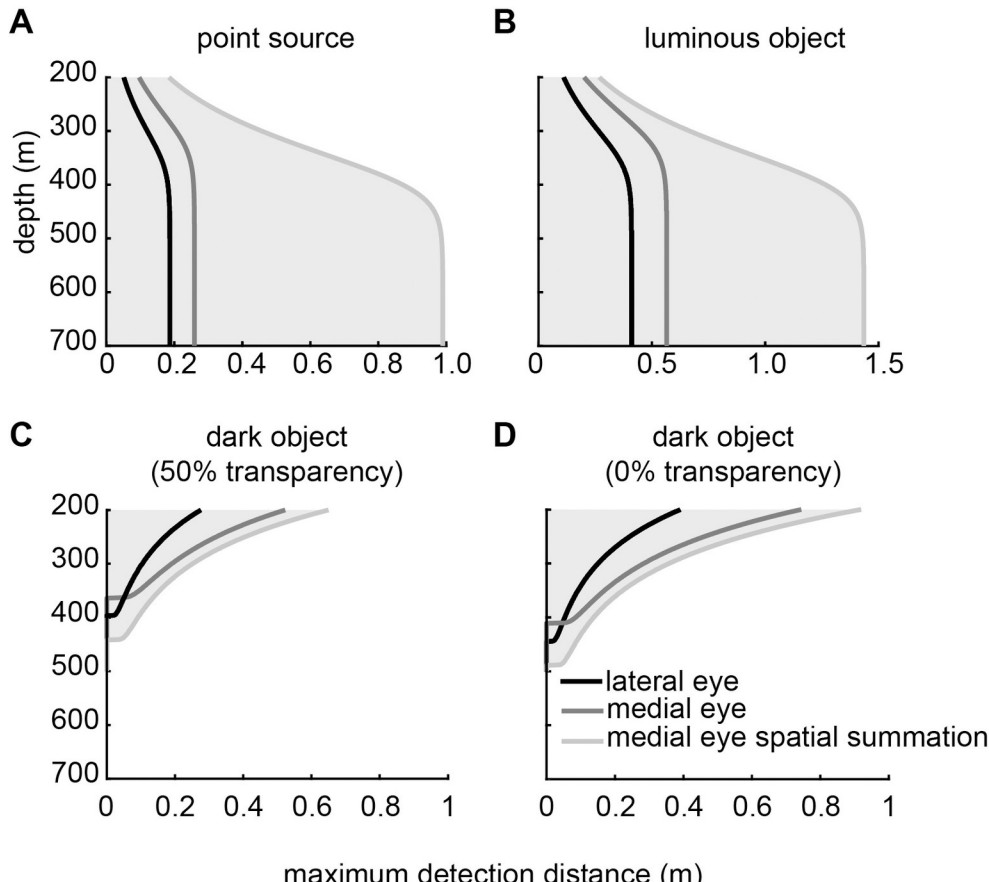

**Fig 8. Comparison of detection abilities between the lateral and medial eyes of *Phronima*.** Model results for the detection distances for (A) the point source, (B) the extended luminous object, and the extended dark object with (C) 50% and (D) 0% transparency. Luminous objects were modelled against horizontal radiance and the dark objects were modelled against downwelling radiance. In all figures we show results for the lateral (black) and medial (dark grey) eyes of *Phronima*, as well as the medial eyes with spatial summation of 19 ommatidia (light grey). Grey shading shows the distances and depths at which detection can occur.

capabilities of the medial eye summating 19 rather than seven ommatidia. At almost all depths, *Phronima*'s medial eyes (which oversample) outperform their lateral eyes (which have optimal sampling) for all targets, even without spatial summation (Fig 8). Without spatial summation, the medial eyes could see the point source 38% further than the lateral eyes against horizontal radiance at 450 m, but if we incorporated spatial summation this improvement increased to 430% (Fig 8A). Similar improvements are seen for the extended luminous object viewed against horizontal radiance, with a maximum increase in detection distance below 450 m depth of 34% without spatial summation and 207% with spatial summation (Fig 8B). Against downwelling radiance at 200 m depth, detection distances for the medial eyes compared to the lateral eyes improved by 87% and 90% for dark objects with 50% and 0% transparency respectively. When we incorporated spatial summation this improvement increased to 120% and 135% respectively (Fig 8C and 8D). Without spatial summation, the lateral eyes have an improved functional depth range (down to ~400 m for 50% transparency and ~440 m for 0% transparency) than the medial eyes (down to 370 m and 410 m respectively) for detecting the extended dark objects. With spatial summation, the medial eyes have a significantly deeper depth range (down to ~440 m and ~490m; Fig 8C and 8D).

We further compared the detection abilities of *Phronima*'s medial and lateral eyes for dark objects with different amounts of transparency (Fig 9). At shallower depths, between 200 and 300 m (Fig 9A and 9B), medial eyes outperformed lateral eyes for almost all transparency values. Detection distances of the medial eye were 90% higher than those for the lateral eye for most transparency values. Spatial summation provided an additional increase of 45% for the medial eye. However, at very high transparency values the lateral eye outperformed the medial eye, unless the medial eye employed spatial summation. This effect increased with greater depths (Fig 9). However, with spatial summation, the medial eye outperformed the lateral eyes for all transparency values. At 400 m depth, the medial eye without spatial summation became almost dysfunctional while the lateral eye was still able to detect objects with 47% transparency at a distance of 2 cm, and the medial eye with spatial summation could detect a 75% transparent object at a distance of 4 cm.

## Discussion

Our model for apposition compound eyes showed that decreasing the acceptance angles of ommatidia resulted in greater detection distances for most targets and most depths, but at the cost of decreased depth range over which vision is useful (Fig 5H and 5I). Our model also showed that spatial summation improved detection distances for all targets, including bioluminescent point sources (Fig 6), as long as receptive fields of neighbouring ommatidia overlapped substantially. When applied to the eyes of *Phronima*, our model showed that both medial and lateral eyes could detect all target types (Figs 8 and 9), but the medial eyes were always better than the lateral eyes, as long as they spatially summate.

### Optical strategies to improve detection ability

In line with Nilsson et al. [9], our results showed that the facet diameter (equivalent to pupil diameter in the models of Nilsson et al. [9]) had the greatest effect on detection distance (Fig 5A–5C). Contradictory to the findings of Nilsson et al. [9], however, we showed that the receptive field size (acceptance angle) had a considerable effect on the detection distance of both point sources and extended objects (Fig 5G–5I). Nilsson et al. [9] found that larger receptor diameters of single lens eyes (which together with the focal length set the receptive field size of retinal photoreceptors in their study) decreased 1) the detection distance of point sources, but had 2) insignificant effects on their ability to detect extended objects. The difference in our results is driven by the fact that Nilsson et al. [9] assumed that receptors dynamically pool their responses according to the angular size of the target. This, by definition, means that increasing or decreasing the receptor diameter has a negligible effect on the detection of extended targets, but decreases detection distances for point sources.

The ommatidial acceptance angle in our model was set by the rhabdom width and the focal length of the ommatidia and ultimately set the receptive field size of individual ommatidia or a channel comprising multiple ommatidia. Fixing the receptive field size to single ommatidia or groups of ommatidia seemed more appropriate for eyes with few ommatidia. We are not aware of any evidence that these animals were able to dynamically adjust their effective channel sizes. The effects of fixing the receptive field size are particularly strong at small distances or for large objects and therefore at the edge of the distribution of useful vision for a particular eye.

Increasing the acceptance angles of the ommatidia improved the range of depths over which vision could be used but decreased detection distances at other depths (Fig 5H and 5I). This was true for both extended luminous and dark objects. A target channel with a larger receptive field could detect extended luminous objects at shallower depths (Fig 5H) and dark

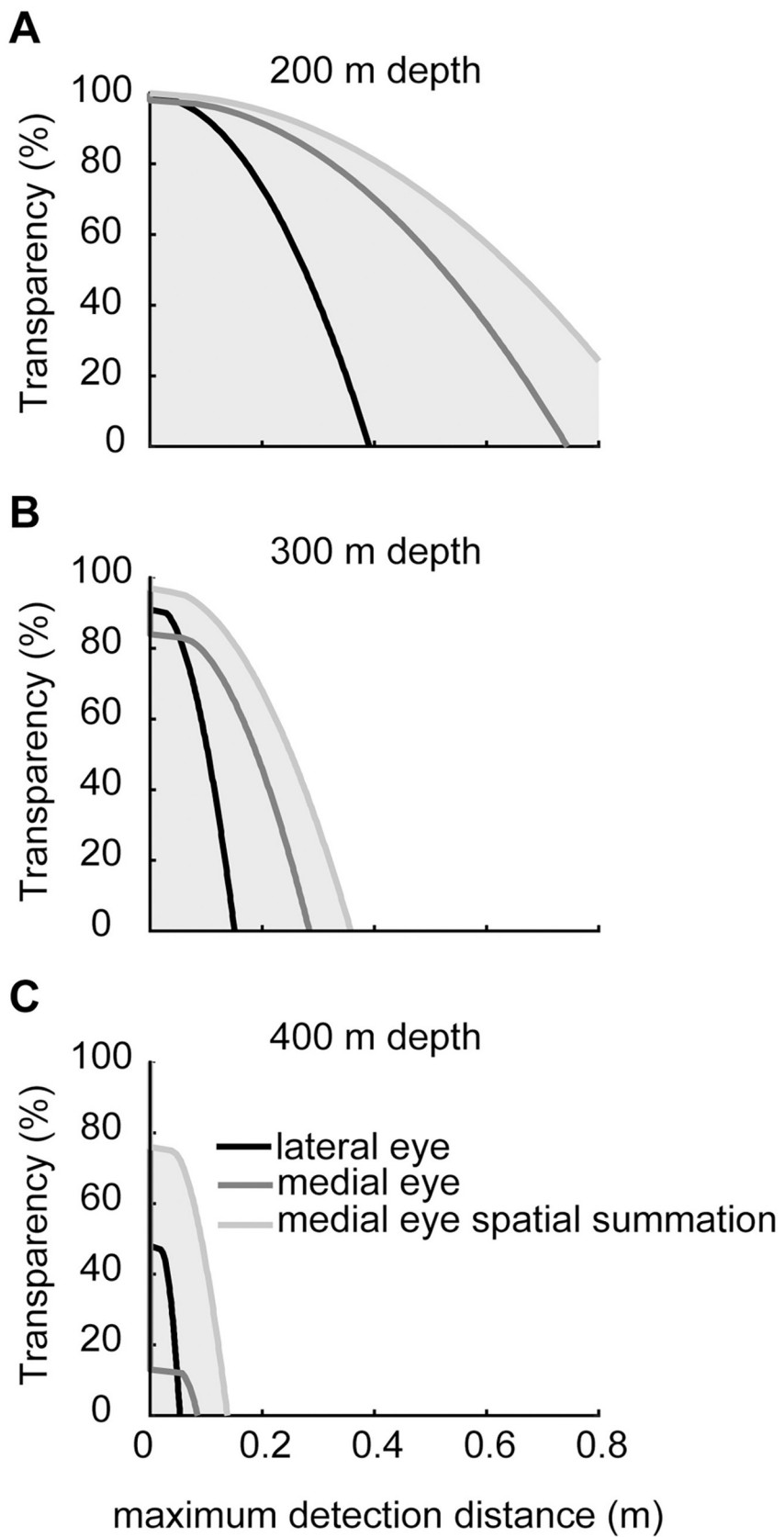

**Fig 9. With increasing depth, more transparent objects become more difficult to detect.** Model results for the lateral and medial eyes of *Phronima* for detection distances of extended dark objects with different transparencies at (A) 200 m (B) 300 m and (C) 400 m depth. Objects were modelled against downwelling radiance. In all figures we show results for the lateral (black) and medial (dark grey) eyes of *Phronima*, as well as the medial eyes with spatial summation of 19 ommatidia (light grey). Grey shading shows the distances and transparencies at which detection can occur.

objects at deeper depths (Figs 5I and S2). At these depths, objects could only be seen at close distances due to the poor contrast, therefore a bigger receptive field could be used to measure the object and background without compromising the contrast. Only when the objects were farther away and their angular size was smaller than the receptive field, their effect on the target channel diluted by the background. Therefore, in the middle of the depth range, where objects were seen at longer distances, smaller acceptance angles improve detection distances because the objects occupied a larger proportion of the target channel. Our results identified a trade-off between depth range and detection distances. Larger acceptance angles allowed animals to increase their depth range, but at the cost of shorter overall detection distances. At the depth limits of useful vision, it was better to see something at very short distances than nothing at all. We therefore predict that animals with larger depth ranges will have larger acceptance angles compared to animals with narrower depth ranges.

## Neural strategies to improve detection ability

**Integration time and quantum efficiency.** Our results showed that increasing the integration time, just like increasing the quantum efficiency of photoreceptors, improved the maximum detection distance of all target types (Fig 5D–5F) in agreement with previous results [9]. However, unlike increasing quantum efficiency, increasing integration time came at a cost: an animal with long integration times sacrificed its ability to resolve moving objects, which has considerable consequences for fast moving animals or animals that need to detect fast moving objects. Therefore, considering our results suggest quantum efficiency has the same effect on detection distance as integration time, increasing quantum efficiency would be a more efficient strategy to improve detection ability.

Both integration time and quantum efficiency are poorly known in most mesopelagic animals. Given the considerable effect that both properties have on the detectability of pelagic targets, it is important that accurate values are obtained, pointing to a need for further work to quantify these properties across a large range of mesopelagic animals. The integration time we have used for *Phronima* (37 ms) was measured by [31], but the quantum efficiency used in our models comes from the shore crab *Leptograpsus* [30], and therefore our absolute detection distance estimates for *Phronima* should be interpreted with this qualification in mind.

**Spatial summation.** We have shown that spatial summation can greatly improve the detectability of targets, depending on the amount of overlap (measured as the ratio of ommatidial acceptance angle to interommatidial angle) between neighbouring ommatidia that summate their receptive fields (Figs 6 and 7). Spatial summation essentially increases the receptive field by summing the receptive fields of neighbouring ommatidia and therefore shows the same depth and detection distance trade-off discussed above for acceptance angle (Fig 6A–6C). However, by increasing the overlap of receptive fields, the increase in the size of the summated receptive field is reduced and the detection distances for all targets are increased (Fig 6D–6F), particularly for bioluminescent bright targets, whose detection is highly sensitive to receptive field size [1].

The overlap of the receptive fields of neighbouring ommatidia increases detection distances but comes at the cost of narrowing the animal's visual field (Fig 7). We did not find any

optimum overlap of the receptive field for either detection distances or search volume (Figs 7 and S3). The optimum solution most likely depends on species specific factors such as the cost functions for search distance and size of visual field, or what detection distances are relevant for animals of a certain size and speed. However, these factors are as yet unknown and our modelling did not provide any firm predictions. The ratio of ommatidial acceptance angle to interommatidial angle from our morphological analysis was 2.4 while previous results reported a value of 9.2 [6]. In our study, interommatidial angles were estimated by averaging the interommatidial angles between a focal ommatidia and its six neighbours, measured across the whole eye using the method described in Bagheri et al. [22]. We estimated acceptance angles by measurements obtained from micro-CT reconstruction. Land [6] estimated interommatidial and acceptance angles by measuring the movement of the pseudopupil across the eye while rotating the animal known degrees [40], or using a photograph of the pseudopupil scanned with a densitometer [10] resulting in an estimated overlap of 7.9. It is possible that the method used in our study underestimates the acceptance angle as it does not account for slightly off-axis rays that may be reflected into the rhabdom from the walls of the cone [10]. Future studies should examine acceptance angles using the most accurate methods available, ideally using intracellular electrophysiology that provides measurement of spatial resolution as truly seen by neurons.

*Phronima*'s medial eyes also have extensive binocular overlap with each other [10]. Binocular overlap is known to increase the probability of detection through probability summation [41]. However, the effect of binocularity was not considered here, but, if employed, would slightly increase detection distances.

## Vision in *Phronima sedentaria*

Compared to *Phronima*'s lateral eyes, the medial eyes incorporate every optical strategy discussed above to increase the detection distances of pelagic targets. The medial eyes have larger facet diameters (156.7 μm compared to 112.9 μm), smaller acceptance angles (3.9˚ compared to 10.2˚) and greater ommatidial receptive field overlap (2.4 compared to 1.0; Table 2) than the lateral eyes. Together, these results strongly suggest that the medial eyes should be better suited to the task of detecting all targets compared to the lateral eyes. Based on our comprehensive anatomical data and new model, we have shown that the medial eyes outperform the lateral eyes in almost every detection task at every depth examined (Figs 8 and 9).

While we do not know whether *Phronima*'s medial eyes employ spatial summation, spatial summation has previously been suggested as a good explanation for the large amount of overlap between neighbouring ommatidia [10]. Spatial summation in compound eyes is thought to occur at the level of the lamina, the most distal neuropil of the arthropod visual system. Supporting this hypothesis, the dendrites of lamina monopolar cells extend into several neighbouring cartridges (the two dimensionally arrayed structural units of the lamina) in the lamina of the nocturnal hawkmoth, to connect the projections of the retinal axons from a single ommatidium [42–45]. This suggests that information can be shared between cartridges/ommatidia, making spatial summation possible. While spatial summation has not yet been shown in crustaceans with apposition compound eyes, behavioural evidence from the nocturnal ghost crab *Ocypode ceratophthalmus*, which has apposition compound eyes, suggests that spatial summation is employed at low light intensities [30]. Fergus et al. [46] suggested that a hyperiid amphipod *Paraphronima gracilis* might use its unique discontinuous retinal configuration paired with neural adaptations for spatial summation via rows of ommatidia instead of within neighbourhoods. Furthermore, Land [10] found an extensive lateral plexus between the retina and the lamina in the medial eyes, but not the lateral eyes, of *Phronima* and suggested that the lateral plexus may be the anatomical basis for spatial summation in the medial eyes.

Whether the lateral plexus between the retina and lamina of the medial eyes contains lamina monopolar cells with wide spreading dendrites, as were found in nocturnal insects and whether such cells are absent from the lateral eyes remains to be seen.

Here, we show that both the medial and lateral eyes can detect all mesopelagic target types across a broad depth range (Figs 8 and 9). But that if the medial eyes employ spatial summation by virtue of overlap between neighbouring ommatidia, they would be better at detecting small, low contrast targets compared to the lateral eyes. Our result suggests that *Phronima*'s medial eyes would be useful for a range of visual tasks, not only the upward-looking detection of dark objects, but also visual tasks involving sight lines in all other directions. The trade-off for having this superior detection ability is that the field of view of the medial eyes is necessarily small, 10–30˚ horizontally and 10–30˚ vertically [6, 10]. However, the habit of living inside a barrel already restricts their field of view to the opening of the barrel. The restriction of the viewing field size in the medial eyes for increased detection distances is, therefore, not a loss to them. The lateral eyes likely provide low resolution vision over a broad field of view ensuring *Phronima* can see in almost every direction while inside and outside of their barrel but still maintain the enhanced ability to detect small, low contrast targets with their medial eyes in a narrow area. Perhaps *Phronima*'s extremely asymmetric double eyes have co-evolved with this animal's unusual behaviour of living inside a barrel formed from gelatinous zooplankton.

## Transparency and detection distances

Like previous results, our calculations show sighting distance and detectability of semi-transparent objects vary with depth. A semi-transparent animal which is visible near the surface may become invisible at greater depths [32] as contrast between background and object decreases. Our results further suggested that transparency, like depth, forces animals to make a trade-off between maximising detection distance versus the ability to see transparent objects; or in the case of depth, maximising detection distance versus the depth range of useful vision. This is not surprising because, similar to depth, transparency also affects object contrast and therefore the effect of acceptance angle on the ability to detect targets at different distances.

Many animals in the deep sea use transparency as a form of camouflage [13]. Therefore, detection of a semi-transparent predator or prey is a significant selective pressure for mesopelagic, vision-dependant animals. For both predators and prey, success greatly depends on their ability to see objects at a large enough distance to react. For prey, a short sighting distance reduces the probability of a successful escape, while for predators, a short sighting distance reduces their chances of successful capture. While most eyes would face a trade-off between smaller detection distances at average depth in exchange for a larger depth range (Figs 8 and 9), the overlapping receptive fields of *Phronima*'s medial eye remove this trade-off – as long as they are able to spatially summate.

## Conclusions

Our computational models provide a framework for future assessments of visual performance in apposition compound eyes in the deep ocean. We have uncovered a trade-off between the depth of vision and detection range based on the acceptance angle of the target channels. Large acceptance angles are important for vision across greater depth ranges, but at the cost of shorter detection distances at intermediate depths. In addition, we have shown that spatial summation, as a result of overlapping receptive fields, is a useful strategy for improving detection of all pelagic targets, including bioluminescent ones. We demonstrate that the medial eyes of *Phronima* are better adapted for detecting all categories of pelagic visual targets, regardless of the direction of view, compared to the lateral eyes. Finally, we hypothesize that the evolution

of their unusual eyes may be related to the restricted viewing field created by the narrow aperture of the "barrel" they often live in.

## Supporting information

**S1 Fig. The radiance of light in the equatorial Pacific at 1005 hrs seen from different viewing directions at different depths (data from [27]).**
(PDF)

**S2 Fig. The effects of visual parameters on modelled detection distances of the extended dark object with 0% transparency against downwelling radiance across depths.** Each column shows the effects on the maximum detection distance of increasing or decreasing, by a factor of two, the: (A) facet diameter, (B) quantum efficiency or integration time, and (C) acceptance angle of the ommatidium. The thick black solid line in each figure shows the result from the average medial eye parameters taken from our *Phronima* specimens (Table 2). The thinner black lines bounding the shaded areas show the result of decreasing (dark grey) or increasing (light grey) the parameters by a factor of two.
(PDF)

**S3 Fig. The effect of receptive overlap between neighbouring ommatidia on search volume.** (A) Search volume for point source initially decreases but after $\Delta\rho$:$\Delta\varphi$ of three the search volume increases as the overlap increases. (B) The search volume for extended luminous object has the same trend as point source except at depth of 200m where detection distance always increases as the overlap increases. (C) The search volume for extended dark object with 50% transparency always increases with overlap at depths of 300 m and 400 m. At 200 m depth, initially there is a slight decrease in search volume. However, after $\Delta\rho$:$\Delta\varphi$ of three the search volume increases as the overlap increases. The number of ommatidia summating in a single channel was calculated as a function of $\Delta\rho$:$\Delta\phi$ ratio[10]. Search volumes were calculated using the derivations shown in the S3 Appendix. The acceptance angle was taken from the medial eyes of *Phronima*, 3.9˚ (Table 2), and we varied the interommatidial angle (from 0.26 to 3.9˚) to model different $\Delta\rho$:$\Delta\phi$ ratios. Dotted vertical lines show the calculations for the overlap measured in our *Phronima* specimens ($\Delta\rho$:$\Delta\phi$ = 2.4) and dashed vertical lines show the calculations for the overlap measured by Land [6]($\Delta\rho$:$\Delta\phi$ = 9.2).
(PDF)

**S1 Appendix. Derivation of Eq 1.8.**
(PDF)

**S2 Appendix. Derivation of the solid angles of extended dark object and background.**
(PDF)

**S3 Appendix. Derivation of the search volume, total acceptance angle of a single channel, and the size of the full visual field.**
(PDF)

**S1 Code. MATLAB codes for models of detection distances of three different targets.**
(ZIP)

## Acknowledgments

We are grateful to Alison Sweeney for the invitation to participate in a collecting trip aboard the R/V *Hugh R. Sharp*, Brett Gonzalez for collection and fixation of the specimens, and Freya Goetz for staining and Micro-CT scanning the specimens.

## Author Contributions

**Conceptualization:** Zahra M. Bagheri, Anna-Lee Jessop, Julian C. Partridge, Karen J. Osborn, Jan M. Hemmi.

**Formal analysis:** Zahra M. Bagheri, Anna-Lee Jessop.

**Funding acquisition:** Julian C. Partridge, Karen J. Osborn, Jan M. Hemmi.

**Investigation:** Zahra M. Bagheri, Anna-Lee Jessop.

**Methodology:** Zahra M. Bagheri, Anna-Lee Jessop, Julian C. Partridge, Karen J. Osborn, Jan M. Hemmi.

**Software:** Zahra M. Bagheri, Anna-Lee Jessop, Jan M. Hemmi.

**Supervision:** Julian C. Partridge, Karen J. Osborn, Jan M. Hemmi.

**Writing – original draft:** Zahra M. Bagheri, Anna-Lee Jessop.

**Writing – review & editing:** Julian C. Partridge, Karen J. Osborn, Jan M. Hemmi.

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
