## [Decision Letter · Decision Letter 0]

14 Jul 2022

Dear Dr Bagheri,

Thank you very much for submitting your manuscript "A new computational model illuminates the extraordinary eyes of Phronima" for consideration at PLOS Computational Biology.

As with all papers reviewed by the journal, your manuscript was reviewed by members of the editorial board and by several independent reviewers. In light of the reviews (below this email), we would like to invite the resubmission of a significantly-revised version that takes into account the reviewers' comments.

Two expert reviewers have now assessed your manuscript, and as you will see, both are very positive to your study and the adaptation of the established Nilsson et al model for apposition eyes. Nonetheless, both have a number of comments and suggestions for your manuscript that can only improve it. In particular, Reviewer 2 asks for more clarity in the development of the methods (and suggests a strategy) and a wider discussion of your results in terms of how optimal Phronima's eyes are for different visual tasks in the habitats where they live (among other things). With regards to greater clarity of the methods, Reviewer 1 found them to be very clear already, and I tend to agree. However, if there is a possibility to accommodate any of Reviewer 2's suggestions, then please do so (as they will no doubt improve clarity even further), but I will not make this specific suggestion for revision mandatory for publication. Reviewer 2's comments are entirely motivated by the desire to make your study have even higher impact, so their suggestions for a wider discussion will certainly benefit the manuscript. Congratulations on an excellent and highly interesting study!

We cannot make any decision about publication until we have seen the revised manuscript and your response to the reviewers' comments. Your revised manuscript is also likely to be sent to reviewers for further evaluation.

Sincerely,

Eric Warrant

Guest Editor

PLOS Computational Biology

Thomas Serre

Deputy Editor

PLOS Computational Biology

Two expert reviewers have now assessed your manuscript, and as you will see, both are very positive to your study and the adaptation of the established Nilsson et al model for apposition eyes. Nonetheless, both have a number of comments and suggestions for your manuscript that can only improve it. In particular, Reviewer 2 asks for more clarity in the development of the methods (and suggests a strategy) and a wider discussion of your results in terms of how optimal Phronima's eyes are for different visual tasks in the habitats where they live (among other things). With regards to greater clarity of the methods, Reviewer 1 found them to be very clear already, and I tend to agree. However, if there is a possibility to accommodate any of Reviewer 2's suggestions, then please do so (as they will no doubt improve clarity even further), but I will not make this specific suggestion for revision mandatory for publication. Reviewer 2's comments are entirely motivated by the desire to make your study have even higher impact, so their suggestions for a wider discussion will certainly benefit the manuscript. Congratulations on an excellent and highly interesting study!

Reviewer's Responses to Questions

**Comments to the Authors:**

Reviewer #1: The manuscript by Bagheri and colleagues "A new computational model illuminates the extraordinary eyes of Phronima" expands existing computational models for underwater vision developed previously for camera-type eyes (such as those of cephalopods and vertebrates) so that they may be applied to apposition compound eyes (such as those of crustaceans, including hyperiid amphipods like Phronima). These expanded computational models will be helpful to many of us in the field of visual ecology. I also want to note my appreciation for the effort the authors put into describing their methods clearly and thoroughly. After presenting their approach to the computational modelling of underwater vision, the authors apply their methods to the unusual eyes of Phronima. By doing so, they demonstrate how the enormous medial eyes of Phronima may be able to perform a wide range of visual tasks over a wide range of depths by combining ommatidia with wide and overlapping visual fields with a high level of spatial summation (which in this scenario involves the pooling of visual input received by adjacent ommatidia). Overall, I found this to be an interesting, useful, and highly professional paper. I only have minor comments for the authors.

Line 54: I think you want "if they bioluminesce" or "if they emit bioluminescence" or "if they are bioluminescent" here.

Line 79: Effects of the transparency of what? The answer becomes clear later, but I think you'll want to specify "effects of the transparency of extended objects" here.

Line 90: I think "morphologically distinct" would be a better choice here than "spatially distinct".

Line 132: Check descriptions for panels A and B -- I think both are missing a few words.

Methods: Please add a few more notes about where the methods developed by Nilsson et al end and the new methods begin. I appreciate the credit the current authors are giving to past authors, but I'm finding the dividing line between past and present to be a bit opaque.

Figure 2 (and perhaps other figures): Please make sure all panels are cited in the text and that they are cited in the correct order. Unless this isn't a requirement for PLOS? I know it's a requirement for some journals.

Table 1: I very much appreciate the addition of Table 1 to the manuscript. It is quite helpful. Please double-check that all relevant terms and values are included here. I think Si (from Section 1.2) might be missing? Perhaps others are too?

Lines 500-501: I'm not sure how to interpret the units here: "Extended objects were 1 cm m in diameter".

Lines 587-588: If you want to test the hypothesis that animals with broader depth ranges have ommatidia with larger acceptance angles, perhaps compare the eyes of pelagic species to those of benthic species? Diel vertical migration makes me skeptical of published depth ranges for pelagic species, but I think one can make comparisons between pelagic and benthic species with a high degree of confidence.

Discussion (overall): It is curious to me that the authors do not cite or discuss the 2015 paper on the eyes of Paraphronima gracilis by Fergus, Johnsen, and Osborn. It also seems odd that the authors do not reference the 2021 paper by Lin et al on optic lobe organization in hyperiid amphipods. I'm not finding any statements in the Discussion that I want to debate, but I would like to see the Discussion updated to include recent papers on the eyes and brains of hyperiid amphipods, such as those by Fergus et al and Lin et al (there may be more, but these two came to mind immediately). Please note that I am not an author on either of these papers!

Reviewer #2: This is a nice analysis of the always-fascinating eyes of the hyperiid amphipod Phronema. The results are a novel, detailed microCT structure of both the medial and lateral eyes, and an extensive repurposing and adaptation of an existing model of visual sensitivity in the deep sea used to analyze the possible performance of these eyes in a few different visual tasks. The paper itself is a good start, but very specific and needs restructuring. With some additional thought, however, I think they could have a much more widely read and cited piece of work.

The methods section outlining the many different use-cases of the Nilsson model they develop is too long and detailed with too many variables introduced, it makes it hard to tell how many different models there are and the differences between them. This needs a good clarifying overhaul. I suggest outlining an initial use of the basic model in detail, with a new schematic figure that illustrates the spatial relationships between the quantities and variables employed in the model. From there, each new use case can state briefly how the model was tweaked, what any additional underlying assumptions are, and simply state the calculation for sensitivity used downstream. The additional algebra and derivations could then be included in a longer supplementary methods section.

Given the difficulty of working in the midwater, there are necessarily uncertainties about important quantities like integration time and quantum efficiency of these ommatidia. The authors find plausible reference values. Similarly, there is a pretty big disagreement between the degree of angular overlap between ommatidia they measured, and older data from Land.

Given this uncertainty, and the interesting work they did to develop the Nilsson model for apposition compound eyes, I think this would be a far more interesting piece of work if they shifted the emphasis of the paper to explore the optimality of compound eyes for the different visual tasks they identify (point sources, extended bright objects, and extended dark objects). For example: what is the optimum angular overlap between ommatidia for viewing point sources at the peak of the organism's light/depth distribution? And how different is that from either their or Land's observations? And are those observations more similar than different given the size and shape of the optimality landscape? Given the shape of this landscape, is Phronema more likely to be a visual specialist or generalist? I think this approach is more likely to provide insight about the specific evolved functions of Phronema's eyes than the current approach of a parameter sweep in the face of significant parameter uncertainty.

Another question that I am curious about. The Nilsson model relies heavily on the assumption of a Gaussian spatial sensitivity to incoming light for a single lens. This assumption was made in the context of camera-type eyes. The optics of the crystallin cones at the surface of Phronema ommatidia are very different. Is this assumption of Gaussian sensitivity profile still usable? The authors don't discuss this point. Is the Gaussian assumption appropriate here, or contrarily, is there any possible sensitivity distribution that would significantly change the model's results?

This manuscript is a nice start on a nice piece of work - I think considering some of these possibilities will both make the work more appropriate for a Computational Biology journal and make the work more sophisticated and more widely read.

**Have the authors made all data and (if applicable) computational code underlying the findings in their manuscript fully available?**

Reviewer #1: Yes

Reviewer #2: None

PLOS authors have the option to publish the peer review history of their article (what does this mean?). If published, this will include your full peer review and any attached files.

Reviewer #1: No

Reviewer #2: No
---

## [Editor Report · Decision Letter 1]

5 Sep 2022

Dear Dr Bagheri,

We are pleased to inform you that your manuscript 'A new computational model illuminates the extraordinary eyes of Phronima' has been provisionally accepted for publication in PLOS Computational Biology.

Best regards,

Eric Warrant

Guest Editor

PLOS Computational Biology

Thomas Serre

Section Editor

PLOS Computational Biology

Thank you for your very thorough and well considered responses to the two reviewers. Your improvements to the manuscript have increased its value and impact, and I now consider it ready for publication.

---

## [Editor Report · Acceptance letter]

16 Sep 2022

PCOMPBIOL-D-22-00769R1 

A new computational model illuminates the extraordinary eyes of Phronima

Dear Dr Bagheri,

I am pleased to inform you that your manuscript has been formally accepted for publication in PLOS Computational Biology. Your manuscript is now with our production department and you will be notified of the publication date in due course.

With kind regards,

Agnes Pap
